

**Assessing the drought resilience of different land management scenarios using a tracer-aided**
**ecohydrological model with variable root uptake distributions**
Cong Jiang [1,2,3], Doerthe Tetzlaff [1,2,3], Songjun Wu[1], Christian Birkel [1,4], Hjalmar Laudon [5], Chris Soulsby [1,3,5,6]
[1] Leibniz-Institute of Freshwater Ecology and Inland Fisheries (IGB), Department of Ecohydrology and
Biogeochemistry, Berlin, Germany
[2] Department of Geography, Humboldt University Berlin, Berlin, Germany
[3] Northern Rivers Institute, University of Aberdeen, Aberdeen, UK
[4] Department of Geography, University of Costa Rica, San Pedro, Costa Rica
[5] Department of Forest Ecology and Management, Swedish University of Agricultural Science (SLU), Sweden
[6] Chair of Water Resources Management and Modeling of Hydrosystems, Technical University Berlin, Berlin,
Germany
*Correspondence to*: Cong Jiang (cong.jiang@igb-berlin.de)
**Abstract.** Land use strongly influences water partitioning, availability, and ecohydrological resilience in drought-
sensitive regions. Forest management plays a critical role through its effects on water use, which depends on
species composition, stand density, rooting depth, canopy structure, and age. However, the ecohydrological
consequences of different forest management strategies—particularly in terms of blue and green water fluxes—
remain poorly quantified for land use planning. This study conducted a series of modelling experiments using the
tracer-aided conceptual ecohydrological model EcoPlot-iso as a decision-support tool. We investigated how
variations in forest type (e.g., broadleaf vs. conifer), density, and root distribution influence water partitioning and
ecohydrological resilience under different wetness conditions in the drought-sensitive lowland Demnitzer
Millcreek catchment (DMC), northeastern Germany. Baseline simulations (2000–2024) across several land use
types were used to develop a reference forest for comparison with alternative forest management scenarios. A key
innovation in this version of EcoPlot-iso was the integration of a depth-dependent root water uptake function,
allowing simulation of transpiration across forests with different rooting distribution, stand ages, and species
compositions. The model was calibrated and validated using seven years of soil moisture and three years of soil
water isotope ($\delta^2$H) data through a multi-criteria approach. Results showed that, on average, evapotranspiration
was 8% higher under conifers than broadleaf forests, and 12% higher than agroforestry. Agroforestry, in contrast,
provided the highest groundwater recharge—11% and 4% more than conifers and broadleaf forests, respectively.
Significant differences in water partitioning between dry and wet years were observed across management
scenarios. Our findings highlight the potential of agroforestry, such as crop–tree mixtures, to mitigate drought
impacts. The modelling framework provides a means to quantify and visualise the effects of land use change on
water availability, supporting more informed decision-making for resilient land and water management.
**1 Introduction**
Land use plays a crucial role in regulating water, carbon, energy, and nutrient cycles by mediating ecohydrological
fluxes and soil water storage dynamics which link interactions between the atmosphere, soils, vegetation and
biogeochemical processes (Mahmood et al., 2014; Pielke et al., 2011; Smith et al., 2021; Sterling et al., 2013).
Among the different types of land cover, forests are particularly important elements of the land use mosaic,
providing a range of ecosystem services, including enhancing infiltration, stabilizing soils, storing carbon,
supplying timber and fuelwood, as well as buffering extreme climate events (Bonan, 2008). However, there are



clear trade-offs, as forests and trees also tend to use more water than contrasting land uses (Bosch & Hewlett,
1982; Calder, 1998). This is because their high Leaf Area Index (LAI) and canopy storage capacities often result
in great interception losses and canopy evaporation, while their deep and dense rooting networks can sustain
transpiration when top soils dry out (Wang-Erlandsson et al., 2014). Consequently, forest management decisions,
(e.g., afforestation, thinning, species selection etc.) can significantly affect water yield, the partitioning into blue
(runoff, groundwater recharge) and green (evapotranspiration) water fluxes, and overall drought resilience
(Falkenmark & Rockström, 2006; Neill et al., 2021).
Sustainable land management also requires consideration of sensitivity to climate change, which is altering
hydroclimatic regimes by shifting precipitation patterns, intensifying drought frequency and duration in many
areas (Huntington, 2006; Trenberth, 2011). These changes can increase atmospheric demand and evaporative
losses, reducing groundwater recharge and surface water availability, and thus exacerbating water scarcity in many
regions (Ault, 2020; Yuan et al., 2023). As land use practices—particularly forest management—strongly
influence water partitioning, understanding their impacts under changing hydroclimatic conditions is essential for
maintaining resilient water and land systems, especially in drought-prone areas.
The understandings on how land use change affects runoff generation, soil moisture storage and
evapotranspiration dynamics have been gradually developed through decades of research, including long-term
experimental watershed studies such as paired catchment experiments on water yield (Bosch & Hewlett, 1982;
Brown et al., 2005, 2013; Hibbert, 1967). However, quantifying the impact of forest management on water
partitioning remains challenging (Guswa et al., 2020). This is due to the complex interplay of climate conditions,
soil properties, vegetation type, and topography, and the difficulty in distinguishing individual ET components
(Kool et al., 2014; Smith et al., 2021; Zhang et al., 2001). These challenges are further compounded by scarce
long-term observational data for forest ecosystems, which are essential given their slow dynamics and lengthy
growth cycles (Tetzlaff et al., 2017). In forest ecosystems, ET is particularly difficult to simulate due to complex
interactions among canopy structure, stomatal behavior, and root water uptake (Tague & Band, 2004). Although
many ecohydrological models include some form of root water uptake conceptualization (e.g., mHM, EcH2O-
iso), the dynamic and species-specific nature of root distribution and function is usually inadequately represented.
This limits the ability of models to fully capture the effects of forest age, species composition, and management
practices on transpiration and soil–plant water fluxes (Dubbert et al., 2023; Kumar et al., 2015). Recent isotope-
based studies (e.g., Knighton et al. (2020)) have advanced understanding of root water uptake (RWU) dynamics,
yet key knowledge gaps remain, including spatiotemporal variability and species-specific uptake strategies
(Knighton et al., 2024). Moreover, traditional hydrological models often struggle to separate evaporation and
transpiration, limiting their ability to accurately simulate the long-term effects of land management on water
partitioning, vegetation dynamics and soil water storage (Birkel et al., 2025) Additionally, while complex climate
and land surface models offer detailed representations by coupling multiple biophysical processes, they often
require extensive computational resources and dense parameterization resulting in high uncertainty (Ricci et al.,
2020). Despite their complexity, many sophisticated models are difficult to effectively calibrate, leading to higher
parametric uncertainty—particularly in data-scarce regions (e.g., Fatichi et al. 2012; Tague & Band 2004). As a
result, these models may have limited capacity to accurately represent the long-term water partitioning dynamics
and subtle ecohydrological feedbacks associated with forest structure, root water uptake, and land management.



This highlights the need for complementary simpler, systematic, long-term modeling approaches that integrate
realistic forest management scenarios to better represent water partitioning and ecohydrological resilience.
Tracer-based ecohydrological modelling offers a promising approach to address these challenges by improving
the characterization of water movement, mixing and storage dynamics under different land cover types (Landgraf
et al., 2023; Luo et al., 2024). Stable water isotopes serve as natural tracers and offer unique isotopic fingerprints
that can differentiate between evaporation and transpiration. This distinction is essential for refining our
understanding of ecosystem water use and for better quantifying the timing and magnitude of water fluxes and
storage. These models, which integrate climatic inputs, vegetation, water and soil dynamics, can facilitate more
robust predictions of ecohydrological responses to land use change and management. However, while complex
process-based, tracer-aided ecohydrological models, e.g. EcH2O-iso (Kuppel et al., 2018; Wu et al., 2023),
incorporate vegetation dynamic modules that enhance process representation, they tend to be highly parameterized,
computationally demanding and require extensive input data (Douinot et al., 2019). In contrast, tracer-aided model
of more intermediate complexity, such as the conceptual, tracer-aided model EcoPlot-iso (Landgraf et al., 2023;
Stevenson et al., 2023) provides a simplified modelling tool that has been shown to provide a robust process-
based framework quantifying the effects of land use on water partitioning (Birkel et al., 2024, 2025).
In this study, we apply the tracer-aided conceptual model EcoPlot-iso to assess how land use – specifically forest
management strategies - influences water partitioning and soil moisture storage in the drought-sensitive, lowland
Demnitzer Millcreek catchment, NE Germany. The catchment is typical of large areas in central Europe where
freely draining, sandy soils combine with a relatively dry and warm climate to limit water availability. To improve
the quantification of transpiration, we introduce a novel development in EcoPlot-iso by integrating a depth-
dependent root water withdrawal function into the transpiration equation. The model is dual-calibrated and
validated using seven years of soil moisture data and three years of soil water isotope data. A series of generic
forest management scenarios—varying in forest density, canopy structure (deciduous, coniferous, agroforestry),
and rooting characteristics—are developed to explore their impacts on vertical water fluxes and ecohydrological
resilience.
This study aims to answer the following research questions:
➢  How does vegetation cover influence water use and partitioning under varying wetness conditions
in a drought-sensitive, lowland catchment?
➢  What are the implications of alternative generic forest management scenarios for water availability
and overall ecohydrological resilience?
➢  How can we optimize the land management strategies to mitigate drought impacts and enhance
ecohydrological resilience in the face of climate change?
**2 Study area**
**2.1 Demnitzer Millcreek catchment (DMC)**
The Demnitzer Millcreek catchment (DMC) is a 66 km² lowland basin (30–90 m elevation) in the State of
Brandenburg, Germany, approximately 55 km east of Berlin (52°23′ N, 14°15′ E) (Figure 1). Located in the



Northern European Plain, it is part of a drought-sensitive region that provides many essential ecosystem services,
including agriculture, timber production, and water supply.
The DMC landscape is dominated by non-irrigated farmland, mostly arable crops and some grazing on more
water-retentive soils brown and gley soils respectively which cover 60% of the catchment in the  (Fig. 1a and b).
Forests cover 36% of the catchment, and include coniferous, broadleaf, and mixed stands. Small urban settlements
(2%) are scattered throughout the catchment, with wetlands on peat soils primarily found along streams in the
central part of the catchment. The climate is temperate with warm summers, with a mean annual temperature of
9.6°C and average precipitation of approximately 558 mm, based on weather station data from 2000 to 2024 (see
Table 2). Potential evapotranspiration (PET) ranges from 584 to 789 mm per year from 2000 to 2024, based on
calculations from this study (see Table 2). Interannual variability in precipitation, including the identification of
dry and wet years, is shown in Figure S3, which highlights deviations from the long-term mean and helps
contextualize recent drought impacts. Rainfall peaks in summer, accompanied by intense convective storms;
however, surface runoff is rare, as the soils are highly permeable and dry in the growing season. Consequently,
the catchment is primarily groundwater-dominated with winter high flows and often dries in the summer (Smith
et al., 2021). The geology consists mainly of glacial and fluvial deposits and base moraines, while the dominant
soil types include poorly drained silty gley brown earth and well-drained podzolic brown earth soils (Figure 1b).
The DMC has a long history of human influence, with significant land use changes affecting its hydrology. In the
18th Century, artificial drainage channels were constructed to convert wetlands into agricultural land. Since the
1990s, efforts in wetland restoration and wildlife recolonization (e.g., beaver recovery) have aimed to enhance
water retention in the landscape. Long-term hydrological and isotopic monitoring (Gelbrecht et al., 1996, 2005;
Smith et al., 2020; Wu et al., 2021) has provided valuable insights into the impacts of agriculture and land use
management on water quality, ecohydrological partitioning and soil water storage. The 2018 European drought
and subsequent prolonged dry periods have exacerbated water scarcity and ecosystem vulnerability (Kleine et al.,
2021). In response, some land owners have explored agroforestry and other adaptive forest and tree management
strategies to improve water retention and landscape resilience (Luo et al., 2024). Agroforestry represents a
transitional system blending low density tree cultivation and with agriculture; either in terms of grazing the
understory vegetation or crops (Landgraf et al., 2022; Quandt et al., 2023). Such systems are characterized by
minimal canopy cover and no artificial irrigation, though mulching is often used to enhance soil moisture storage.
Such systems typically involve rows of small deciduous trees or shrubs (≤2 m in height), spaced 2–3 m apart,
interplanted with rainfed legumes (Landgraf et al., 2023). Given the long-term monitoring record and ongoing
land use change, DMC serves as a useful site for assessing the impacts of changing forest management on water
partitioning, soil moisture and ecohydrological resilience under different wetness conditions.

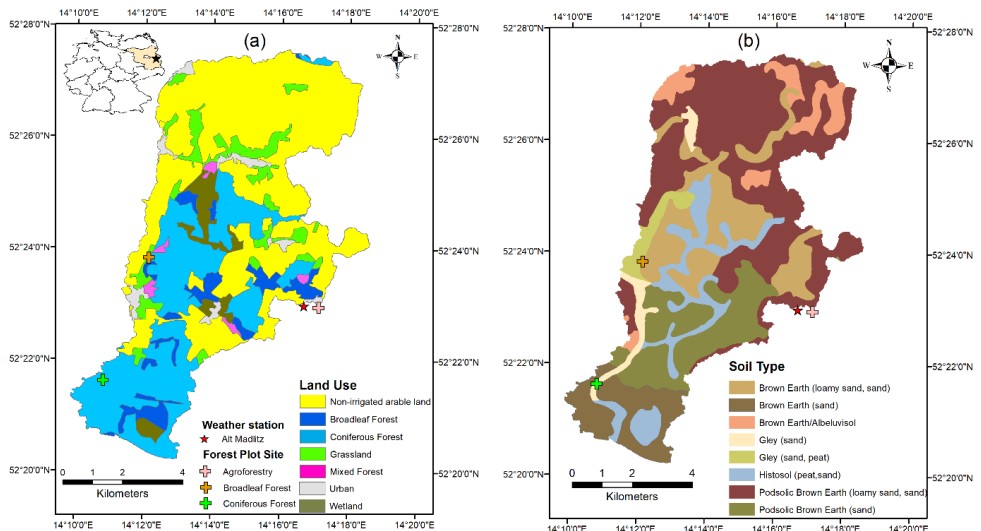

**Figure 1.** Location, land use (a) and soil type (b) map of the Demnitzer Millcreek catchment, showing the
current distribution of broadleaf forests, conifer forests, agroforestry, cropland, and grassland.
**2.2 Forest Plot Site**
To investigate the effects of forest management scenarios on water partitioning and ecohydrological resilience, a
monitoring predominantly broadleaf forest plot site was selected within the drought-sensitive DMC in the NE
Germany. This plot represents a key forest type central to the modelling experiments: a relatively mature (~60
years old) broadleaf forest system. Moreover, it is formed on the extensive freely draining sandy brown soils that
are particularly drought sensitive in DMC due to their poor water retention characteristics. The location is shown
in Figure 1, and site characteristics are described below with more details available in Kleine et al. (2021) and
Landgraf et al. (2023).
Specifically, the broadleaf forest site is dominated by mature European oak (Quercus robur) with a few Scots pine
(Pinus sylvestris) present within the plot. Additional species including Norway maple (Acer platanoides), elm
(Ulmus spp.), and hazel (Corylus avellana) are found within 10 m of the plot boundary. The soil is a freely draining
Lamellic Brunic Arenosol (Humic), characterized by loamy sand to sand textures. This corresponds to a typical
brown earth in regional classification systems.
**Table 1.** Summary observed soil type and soil moisture at three forest sites.

| Site | Soil Type | Texture | Layer | Soil Moisture (mm) | | | |
|---|---|---|---|---|---|---|---|
| | | | | Max | Min | Mean | SD |
| Broadleaf forest | Brown Earth | Loamy sand/sand | 0 to 10 cm | 26.28 | 3.50 | 13.67 | 6.30 |
| | | | 10 to 30 cm | 56.19 | 6.86 | 24.68 | 11.70 |
| | | | 30 to 100 cm | 147.51 | 25.83 | 71.71 | 33.50 |
| | | | 10 to 30 cm | 53.35 | 7.15 | 29.75 | 13.49 |
| | | | 30 to 100 cm | 223.62 | 86.83 | 163.41 | 41.98 |



## 3 Method and Data

### 3.1 Model Framework and Structure

This study employs the EcoPlot-iso model, a tracer-aided ecohydrological modelling framework designed to simulate key ecohydrological and isotopic transformations that characterise water partitioning at the plot scale (Birkel et al., 2024; Landgraf et al., 2023; Stevenson et al., 2023). EcoPlot-iso is a process-based conceptual model that simulates key ecohydrological fluxes, including interception, throughfall, infiltration, preferential flow, surface runoff, percolation, and groundwater recharge, as well as evapotranspiration components such as canopy evaporation, soil evaporation, and transpiration (Figure 2a). These processes are represented within a vertical structure comprising a single canopy layer and three soil layers (0–10 cm, 10–30 cm, and 30–100 cm) (Figure 2b). Recently, the isotope tracking module was further developed to include fractionation and mixing processes, allowing EcoPlot-iso to differentiate evaporation from transpiration and improve water flux estimates. The required input variables (Table 2) include meteorological data such as precipitation, potential evapotranspiration (PET), air temperature, and relative humidity, along with isotopic data (precipitation isotope) and vegetation-related parameters (leaf area index, LAI).

EcoPlot-iso has been applied in diverse climatic and hydrological settings, including a one-year simulation in Scotland (Stevenson et al., 2023), a one-year simulation at the Demnitzer Millcreek (DMC) site in the Northern European Plain (Landgraf et al., 2023), and a four-year simulation in the humid tropics of Costa Rica (Birkel et al., 2024). Building on these applications, this study employs EcoPlot-iso for a long-term tracer-aided ecohydrological simulation to assess the effects of different forest management scenarios on water partitioning and ecohydrological resilience.

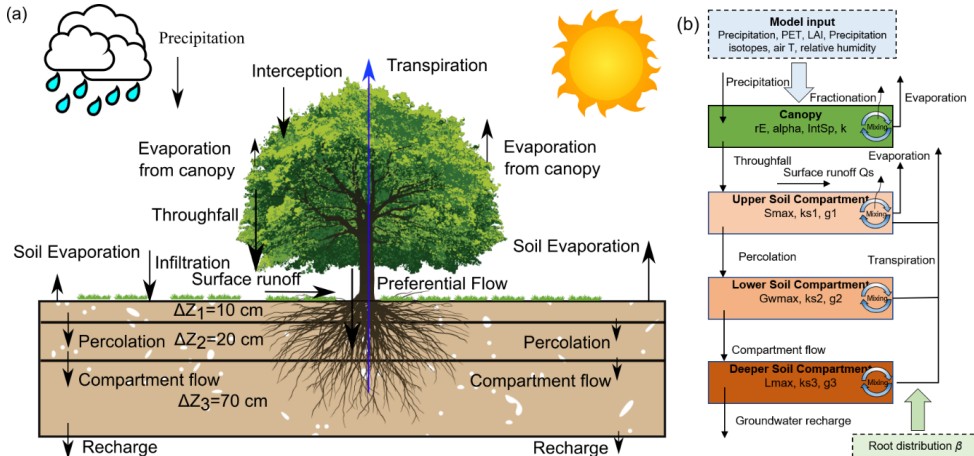

**Figure 2.** (a) Schematic representation of the ecohydrological fluxes and water partitioning in the EcoPlot-iso model illustrating major water fluxes and storage components; (b) Conceptual framework and key parameters of the EcoPlot-iso model(Landgraf et al., 2023; Stevenson et al., 2023), highlighting the key ecohydrological processes simulated in this study.



**3.2 Model Adaptations: Integrating Root Distribution into the Transpiration Equation**

Although root water uptake plays a critical role in soil–plant–atmosphere interactions, it was not explicitly represented in EcoPlot-iso (Stevenson et al., 2023) This study introduces a novel depth-dependent root uptake function to improve the model's simulation of transpiration and water partitioning across different root distributions. This adaptation enables the model to account for variations in rooting depth and water uptake efficiency across land use types—such as young and mature forests or contrasting vegetation covers—that affect soil water extraction. Specifically, a new transpiration equation was implemented to calculate root water uptake across three soil compartments—shallow, middle, and deep—by incorporating depth-specific uptake efficiency:

$$T_{p1} = r_{L1} * \left(T_P - E_i\right) * \left(\frac{STO}{S_{\max}}\right) \tag{1}$$

$$T_{p2} = r_{L2} * \left(T_P - E_i - T_{p1}\right) * \left(\frac{GW}{GW_{\max}}\right) \tag{2}$$

$$T_{p3} = r_{L3} * \left(T_P - E_i - T_{p1} - T_{p2}\right) * \left(\frac{SDeep}{SDeep_{\max}}\right) \tag{3}$$

where $T_{p1}$, $T_{p2}$, $T_{p3}$ represent the transpiration from the upper, lower, and deeper soil compartments, respectively. $E_i$ denotes the canopy evaporation. $STO$, $GW$, $SDeep$ represent the water storage in the upper, lower, and deeper soil compartments. $S_{max}$, $GW_{max}$, $L_{max}$ are the maximum water storage capacities of these compartments. $r_{L1}$, $r_{L2}$ and $r_{L3}$ represent the root water withdrawal efficiency in the upper, lower, and deeper soil compartments, respectively.

To explicitly link root water uptake to soil moisture availability and transpiration demand, an efficiency factor $r(z)$ was introduced. The exponential root water withdrawal efficiency function is defined as:

$$r(z) = e^{-\beta z} \tag{4}$$

where $r(z)$ represents the capacity of roots to extract water at depth z, and $\beta$ is the decay rate, which determines how quickly root activity decreases with increasing depth. A higher $\beta$ value concentrates root activity near the surface, while lower $\beta$ values allow for deeper water uptake (see Supplement Figure S1).

**3.3 Model Setup and Input and Observation Data**

The EcoPlot-iso model was applied to DMC across four sites with different dominant land use: broadleaf forest, cropland, agroforestry, and grassland over a 25-year period (2000–2024) at daily timesteps. Soil moisture initialization was based on observed data, and a one-year spin-up period was included before each simulation to stabilize initial conditions. The input datasets required for the model—climate, vegetation, soil moisture, and isotope data—are summarized in Table 2. Climate variables, including precipitation, temperature, wind speed, and relative humidity, were primarily obtained from the Müncheberg weather station (DWD, German Weather Service, ~20 km from DMC). Potential Evapotranspiration (PET) was calculated using the FAO Penman-Monteith equation, while net radiation was derived from ERA5 reanalysis data (Hersbach et al., 2020). The Leaf



Area Index (LAI) was obtained from the MODIS 8-day resolution dataset and interpolated to daily timesteps. To
improve accuracy and reduce data noise, the MODIS LAI was further adjusted using in-situ LAI measurements
(maximum and minimum values), following Smith et al. 2021 and Wu et al. (2023). The complete set of time
series input data used to drive the EcoPlot-iso simulations in the Demnitzer MillCreek Catchment for 2000-
2004—including daily precipitation, precipitation isotopes ($\delta^2$H), air temperature, relative humidity, Leaf Area
Index (LAI), and potential evapotranspiration (PET)—is presented in Figure S2 of the Supplementary Material.
Surface soil moisture (0–10 cm) was measured using a handheld soil moisture device on a monthly basis during
two periods of more detailed observations in 2018–2019 and in 2021. For subsurface soil moisture, permanently
installed soil moisture probes (two replicates at each depth) were used to continuously monitor Volumetric Water
Content (VWC) at 15-minute intervals at four sites (Figure 1). To facilitate data processing and consistency, all
soil moisture datasets were aggregated into daily mean values, resulting in one VWC value per site and soil depth.
A summary of the measurement devices, depth intervals, and aggregation methods is summarized in Table S1.
Daily precipitation samples for stable isotope analysis from June 2018 onward were collected at the Hasenfelde
AWS, and earlier data were obtained from the Berlin weather station. Soil water isotopes were sampled from bulk
soil at the four plot sites at five depths (0–5, 5–10, 10–20, 20–30, and 30–50 cm) every 3–4 weeks during the
growing season. The isotope data were aggregated according to the thickness of the corresponding model soil
compartments. All isotope values are reported relative to Vienna Standard Mean Ocean Water (VSMOW). Further
details on site instrumentation and data collection are described in Landgraf et al. (2022).
**Table 2.** Summary of the used climate, vegetation, soil moisture, and isotope data

| Data | Unit | Period | Timestep | Acquisition |
|---|---|---|---|---|
| *Climate data* | | | | |
| Precipitation | mm/d | 2000-2024 | Daily | Muencheberg weather station (52.52°, 14.12 °) |
| Temperature | °C | | | |
| Windspeed | m/s | | | |
| Relative humidity | % | | | |
| Net shortwave radiation | W/m$^2$ | | Hourly | ERA5 |
| Net longwave radiation | | | | |
| Potential evapotranspiration | mm/d | | Daily | FAO Penman-Monteith equation |
| *Vegetation data* | | | | |
| Leaf area index | - | 2000-2024 | 8-days | MODIS at broadleaf forest, coniferous, and agroforestry sites |
| *Soil data* | | | | |
| Soil moisture | % | 2018-2024 | Daily | broadleaf forest, cropland, agroforestry, and grassland sites |
| *Isotope data* | | | | |
| Precipitation isotope $\delta^2$H | ‰ | 2000-2024 | Daily | Hasenfelde (52.41°N, 14.19°E), weather station in Berlin |
| Soil water isotope | | 2018-2019, 2021 | Daily | Manually at broadleaf forest, cropland, agroforestry, and grassland sites |




**3.4 Model Calibration and Validation**

The EcoPlot-iso model was calibrated using the Monte Carlo method and a multi-criteria approach based on soil moisture and soil water isotope data for each site. For each model run, a total of 100,000 parameter sets were generated using the Latin Hypercube Sampling (LHS) within a Monte Carlo framework (McKay et al., 1979) to broadly sample the parameter space and capture a wide range of plausible model behaviors. The initial parameter ranges, representing the widest physically feasible values for the site, were determined based on a literature review and site-specific knowledge, with identical constraints applied across all vegetation types.

Model performance was evaluated using the modified Kling-Gupta Efficiency (*mKGE*) (Kling et al., 2012), optimizing the averaged *mKGE* for soil moisture (*mKGE$_{sm}$*) and soil water isotopes (*mKGE$_{iso}$*) across the three soil depth layers (*i*) to ensure robust parameter selection (Eq. 5). Calibration followed a two-step refinement process. In the first step, based on the initial parameter ranges, the top 60th percentile of best-performing simulations—ranked by average *mKGE*—along with their corresponding calibrated parameter sets, were retained. In the second step, the model was re-run using the retained parameter space, and the 100 best simulations were selected from the top 60th percentile to ensure optimal parameter selection. The model parameters, their initial ranges, and the refined ranges for each of land use are summarized in Table S2 in the Supplement.

$$mKGE = \frac{\sum_i^3 mKGE_{sm} + \sum_i^3 mKGE_{iso}}{6} \tag{5}$$

**3.5 Development and Application of a Generic Forest Management Scenario Framework**

To assess the general impacts of different forest management strategies on water partitioning and ecohydrological resilience, we developed a framework for quantifying generic forest management scenarios based on simulations at the broadleaved forest site at DMC. Baseline simulations (2000–2024) were established using EcoPlot-iso at the broadleaf forest site.

From this baseline calibration, we retained the top 100 best-performing simulations—ranked by average modified Kling-Gupta Efficiency (mKGE)—and their corresponding parameter sets. These calibrated parameter sets were then used for scenario testing to ensure robust model performance across all simulations. To isolate the effects of forest characteristics and management, all scenario simulations were driven using the same climate input data and precipitation isotope time series as the baseline, along with consistent forcing data for potential evapotranspiration. Additionally, site-specific Leaf Area Index (LAI) data were adjusted for the three forest types: broadleaf, coniferous, and agroforestry, which were derived from 8-day MODIS remote sensing products (2000–2024) (Table 2 and Figure S2d).

The scenario framework varied three key dimensions of forest management:

a) Forest density was varied by multiplying the reference Leaf Area Index (LAI) by a scaling factor ranging from 0.2 to 1.8. Higher forest density was represented by scaling factors >1.0, indicating denser canopy cover, while lower forest density corresponded to factors <1.0, reflecting more open canopy conditions.

b) Species composition was varied by implementing three canopy types—broadleaf, conifer, and agroforestry—each assigned type-specific LAI values derived from MODIS data corrected using site data at DMC (see Section 3.3) to reflect differences in forest structure and function.





c) Root water uptake efficiency was varied by parameterizing $\beta$ values ranging from 0 to 2.0 to represent vertical root distribution. Lower $\beta$ values indicated deeper rooting systems (e.g., older or deep-rooted species), while higher values represented shallower rooting systems (see Fig. S1).

This generic and scalable framework enables systematic simulation of long-term forest management impacts on water partitioning, soil moisture dynamics, and ecohydrological resilience under consistent climatic conditions. Although EcoPlot-iso was originally developed for plot-scale applications, it is applied here to represent ecohydrological fluxes in a range of well-characterized sites within the DMC region. The model employs a one-dimensional approach that does not explicitly account for lateral fluxes; however, this simplification is intentional. It enables clearer interpretation of process-level dynamics under contrasting vegetation and climate conditions, making it suitable for general scenario analysis. This assumption is especially justified in the DMC catchment, which is characterized by flat, lowland topography and is predominantly governed by vertical hydrological fluxes (Kleine et al., 2021; Smith et al., 2020).

The aim was not to reproduce exact spatial patterns, but to develop a generalizable understanding of how forest structure influences vertical water fluxes and soil moisture. The framework thus serves as a practical tool for assessing broad ecohydrological responses to forest management. Ultimately, the goal was to inform stakeholders of the potential impacts of changes in canopy structure and forest age on long-term water availability and ecohydrological resilience in drought-sensitive lowland catchments.

## 4 Results

### 4.1 Dynamics of Soil Moisture and Soil Water Isotopes at the Broadleaf Forest Site

Figure 3 shows the 25-year baseline simulations of soil moisture and soil water isotopes at the broadleaf forest site. In general, the model effectively captures the magnitude, frequency, extremes, and timing of soil moisture dynamics. Model results show surface soil moisture shows higher variability than deeper layers. Based on the Kling-Gupta Efficiency (KGE), soil moisture simulations appear to perform better in the deep layer than in the shallow and lower layers, though this may partly reflect the more limited variance in deeper soil moisture. In addition, the model slightly overestimates low soil moisture in the deeper layers during wet summers (e.g., 2023, 2024) and underestimates soil moisture during dry winters (e.g., 2021 and 2022). Furthermore, soil water isotope simulations perform well, with better performance in the intermediate layer than in surface and deeper layers in terms of KGE. The uncertainty range of soil water isotope simulations is narrower than that of soil moisture, indicating lower uncertainty in the isotope predictions.

Table 3 shows the Kling-Gupta Efficiency (KGE) values for soil moisture and soil water isotopes across different land use plots. In all other cases the KGEs for soil moisture are similar to the broadleaved plot, and soil water isotopes are reasonably reproduced, indicating the model's robustness and transferability. These results provide strong support for the appropriateness of applying EcoPlot-iso to assess the impacts of alternative forest management scenarios in subsequent analyses.





**Table 3.** Kling-Gupta Efficiency (KGE) values for soil moisture and $\delta^2$H, based on observed values compared to
the mean simulated values.

| Forest sites | Soil moisture | | | Soil water isotope $\delta^2$H | | |
|---|---|---|---|---|---|---|
| | Upper soil compartment | Lower soil compartment | Deep soil compartment | Upper soil compartment | Lower soil compartment | Deep soil compartment |
| Broadleaf Forest | 0.60 | 0.72 | 0.84 | 0.58 | 0.74 | 0.64 |
| Agroforestry | 0.72 | 0.76 | 0.78 | 0.81 | 0.84 | 0.78 |
| Grassland | 0.87 | 0.67 | 0.71 | 0.72 | 0.76 | 0.60 |
| Cropland | 0.53 | 0.54 | 0.71 | 0.82 | 0.84 | 0.28 |

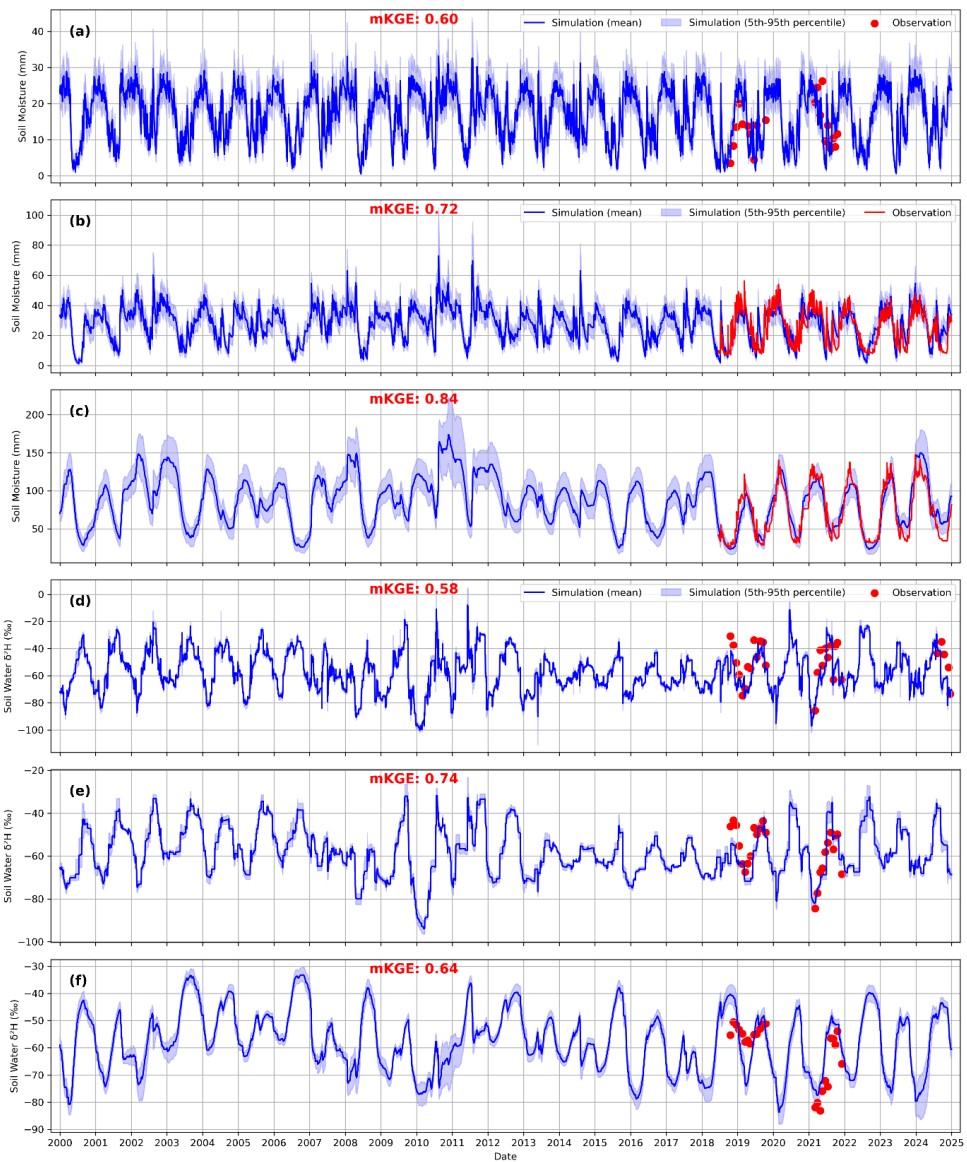

**Figure 3.** Long-term (2000–2024) simulations of soil moisture and soil water isotope (δ²H) at three different depths using EcoPlot-iso at a broadleaf forest site in the Demnitzer Millcreek catchment. (a–c) Simulated (mean ± 5th–95th percentile) and observed soil moisture at surface (0–10 cm), lower (20–30 cm), and deeper (30–100 cm) layers. (d–f) Simulated (mean ± 5th–95th percentile) and observed soil water isotopic composition (δ²H) at corresponding depths. The blue line represents the mean value of the 100 best simulations, while the shaded area indicates the range between the 5th and 95th percentiles of these simulations. The red points and red line represent observed values. Kling-Gupta Efficiency (KGE) values for each simulation are indicated in the respective panels.




### 4.2 Water Balance Components Under Different Wetness Conditions

Figure 4 presents the mean monthly water balance components and their changes between dry and wet years for the baseline simulation at the broadleaved forest site from 2000 to 2024. Groundwater recharge dominates blue water fluxes, while surface runoff is rare and occurs only during extreme summer rainfall events (Figure 4a). Transpiration and canopy evaporation dominate in summer, while soil evaporation peaks in spring. In dry years, recharge declines and dominates the intermonthly variation (Figure 4b), whereas in wet years, it increases following precipitation anomalies (Figure 4c). Despite differences in annual wetness—across both dry and wet years—transpiration remains relatively stable (Figure 4d), indicating resilient vegetation function. This stability likely reflects the mature age of the forest (~60 years), although gradual changes in forest structure over the 20-year period may also play a role. These seasonal patterns offer key insights into water partitioning under broadleaf forest conditions and establish an important baseline for evaluating the impacts of alternative forest management scenarios.

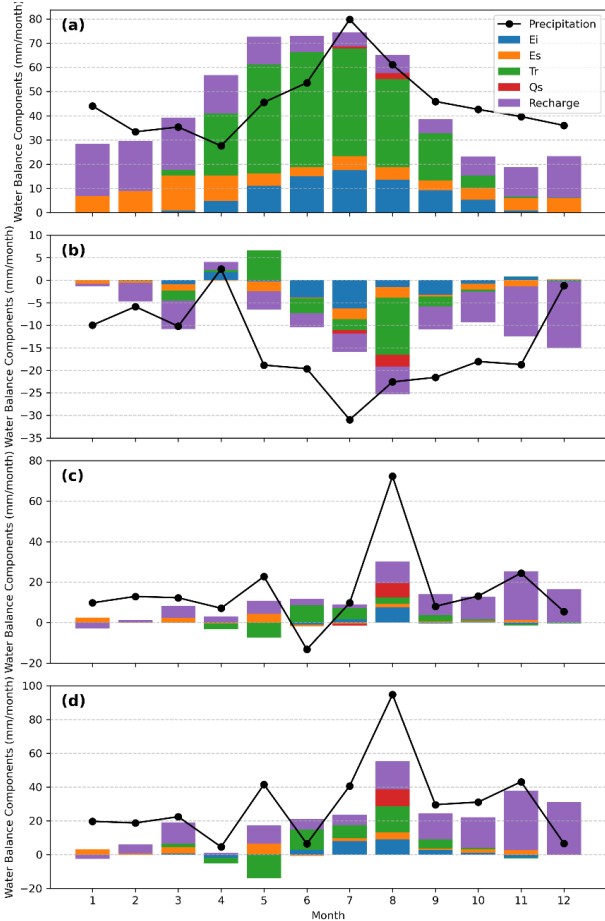

**Figure 4.** Mean monthly water balance components for the period 2000–2024, simulated using EcoPlot-iso for a broadleaf forest site in the Demnitzer Millcreek catchment, based on the mean of the best 100 parameter sets (see Section 3.4 for details). (a) Long-term mean monthly water balance. (b) Deviations of dry years (2006, 2018,





2022) from the long-term mean. (c) Deviations of wet years (2002, 2007, 2010, 2023) from the long-term mean.
(d) Differences between wet and dry years.

### 4.3 Impacts of Forest Management on Water Partitioning and Soil Moisture

### 4.3.1 Water Balance and Partitioning Across Forest Types

Figure 5 compares the mean annual water balance components simulated across broadleaf forest, coniferous forest,
and agroforestry types based on the average of the best 100 simulations. LAI was derived from DMC data for
each forest type, while the LAI scaling factor and root parameters were kept constant across vegetation types.
Results showed that evapotranspiration under coniferous forests accounted for 8% more of annual precipitation
than broadleaved forests, and 13% more than in agroforestry systems. This was primarily due to higher
transpiration (Tr) and canopy interception evaporation (Ei). In contrast, soil evaporation (Es) and groundwater
recharge (Recharge) were lowest in conifers and highest in agroforestry. Agroforestry had 13% more groundwater
recharge relative to annual precipitation than conifers, and 4% more than broadleaf forests. Transpiration
partitioning across root zones (Tr_Upper, Tr_Lower, Tr_Deep) was similar across all forest types, while surface
runoff (Qs) remained minimal and nearly identical. These results reflect the influence of forest structure and
canopy cover on ecohydrological partitioning, with coniferous systems favoring atmospheric losses and
agroforestry promoting soil evaporation and subsurface recharge. They underscore the trade-offs between
evapotranspiration and groundwater recharge across different forest types.

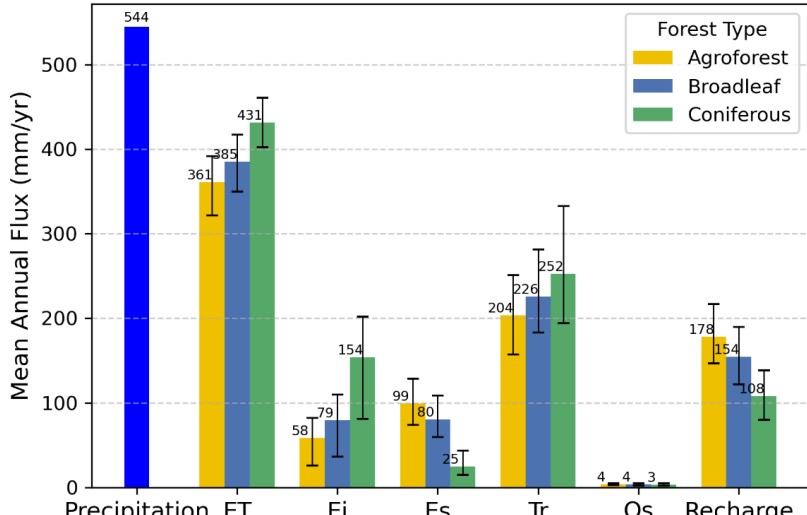

**Figure 5**. Comparison of mean annual water balance components across different forest types: broadleaf (blue),
coniferous (green), and agroforestry (yellow). Bars represent the mean annual flux based on 25-year totals, with
error bars indicating the 5th and 95th percentile ranges of the 100 best simulations. All simulations were conducted
under baseline conditions with a fixed forest root parameter $\beta$ of 0 and LAI scaling factor of 1.0.
Figure 6 presents ternary diagrams illustrating the relative partitioning of key water flux components across three
forest types under baseline conditions. This shows the predominance in transpiration in all three cases (Fig. 6a).
Coniferous forests show a distinct pattern, with the lowest soil evaporation (Es) (Fig. 6a) and groundwater




recharge (Fig. 6b) compared to broadleaf and agroforestry systems. In contrast, broadleaf and agroforestry forests
display largely overlapping partitioning patterns, except for soil evaporation, which differs notably between the
two.

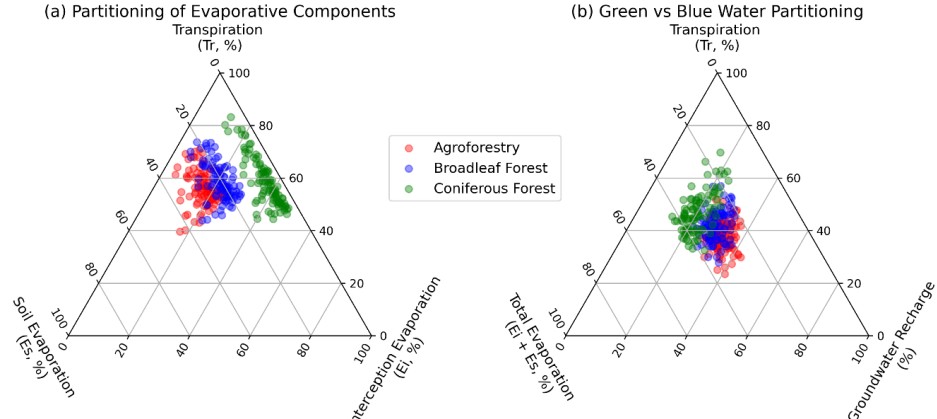

**Figure 6.** Water flux partitioning illustrated using ternary plots based on 100 model simulations for three forest
types: Agroforestry, Broadleaf Forest, and Coniferous Forest, under baseline conditions (root parameter $\beta = 0$,
LAI scaling factor = 1.0). (a) Partitioning of total evapotranspiration into transpiration (Tr), soil evaporation (Es),
and interception evaporation (Ei). (b) Partitioning of water fluxes into green water (Tr and E = Ei + Es) and blue
water (groundwater recharge). Each point represents the normalized annual mean flux from a 25-year simulation.
Colored markers denote different forest types.
**4.3.2 Interannual Patterns and Variability of Water Fluxes**
Figure 7 provides a detailed visualization of the isotope-informed green and blue water partitioning across
different forest management scenarios. The heatmaps present the key ecohydrological fluxes, including
evapotranspiration (ET) (a–c), groundwater recharge (Recharge) (d–f), transpiration (Tr) (g–i), ET partitioning
(ET/P) (j–l), groundwater recharge partitioning (Recharge/P) (m–o) and green water partitioning (Tr/ET) (p–r)
for the three forest types: agroforests, broadleaf forests, and coniferous forests. Evapotranspiration (ET) ranges
from 231 mm/yr to 453 mm/yr across different scenarios, with ET proportion relative to precipitation varying
from 0.42 to 0.83, respectively. In contrast, groundwater recharge ranges from 88 mm/yr to 307 mm/yr.
Transpiration (Tr) varies between 49 mm/yr and 238 mm/yr, with the corresponding green water partitioning
(Tr/ET) ranging from 0.21 to 0.53. These results underscore the significant influence of vegetation type and
structure on ecohydrological fluxes and water partitioning outcomes.
Furthermore, annual mean values show that both transpiration and evapotranspiration increase with higher LAI
scaling factors, while groundwater recharge decreases (Figure 8). Figures 8a and 8b illustrate the trade-off between
increased ET and reduced groundwater recharge under different forest management scenarios. Transpiration and
ET rise rapidly at first, then slow down and transpiration even slightly decreases for conifer forests due to soil
moisture limitation (Figure 8c). This decline is not observed in broadleaf or agroforestry systems, likely due to
their different seasonal LAI patterns. While summer LAI values for broadleaf and coniferous forests may be
similar, the consistently high year-round LAI in conifers can exacerbate moisture stress.



At higher LAI levels, transpiration decreases slightly while interception and evaporation from the canopy increase
(Figure S4). In dense coniferous stands, excessive interception and persistently dry soils limit root water uptake,
reducing vegetation function. This highlights a trade-off between transpiration and interception evaporation. The
resulting moisture limitation suggests that such high-density forests may not be sustainable under water-limited
conditions, as this negative feedback could constrain long-term forest growth and persistence. In addition, forests
with shallow-rooted species—such as young trees—tend to transpire less, generate more groundwater recharge,
and exhibit lower Tr/ET ratios compared to deep-rooted forests. However, even at constant LAI, transpiration
declines with increasing canopy density, suggesting that rooting depth alone cannot compensate for moisture
limitations in dense forests.



**Figure 7.** Green and blue water partitioning across forest types and LAI scaling factors. The heatmaps illustrate evapotranspiration (ET) (a–c), groundwater recharge (d–f), transpiration (Tr) (g–i), ET partitioning (ET/P) (j–l), and green water partitioning (Tr/ET) (m–o) for three forest types (Agroforest, Broadleaf, and Conifer). The x-axis represents scaling factors (forest density), while the y-axis represents root parameters (forest ages). Each heatmap includes numeric values for clarity, with red-outlined cells indicating the baseline simulations (Broadleaf forest, scaling factor = 1, root parameter = 0).

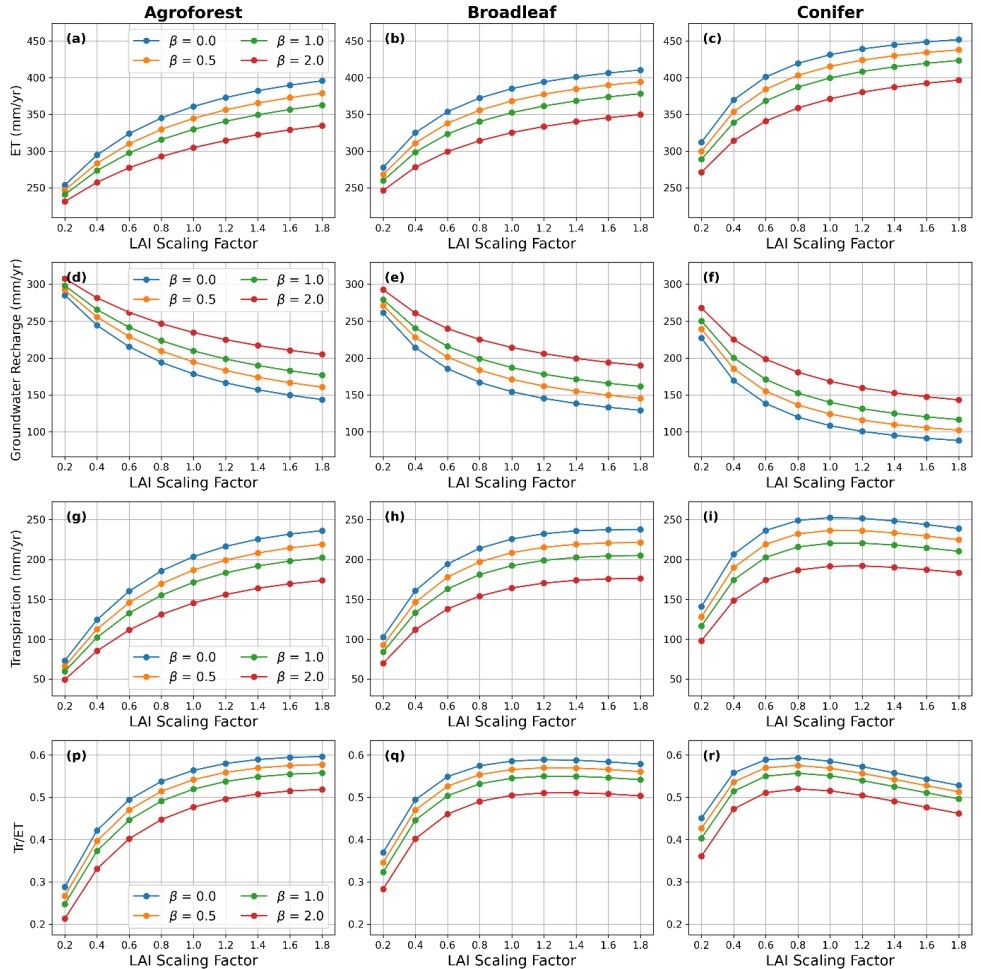

**Figure 8.** Annual mean ecohydrological fluxes for three forest types (Agroforest, Broadleaf, and Conifer) under varying LAI scaling factors and root depth scenarios. Panels (a)–(c) show evapotranspiration (ET), (d)–(f) show groundwater recharge, (g)–(i) show transpiration, and (p)–(r) show the ratio of transpiration to total evapotranspiration (Tr/ET). Each line represents a different forest age class (i.e., root depth) denoted by β values.

### 4.3.3 Seasonal and Monthly Dynamics of Water Fluxes

Figure 9 shows monthly deviations in water balance components under different forest management scenarios, relative to a baseline broadleaf forest. Agroforestry scenarios tend to have lower transpiration and canopy evaporation, but higher soil evaporation during summer (Fig. 9a). They are also associated with greater groundwater recharge from summer through the following winter. A shift from broadleaf to conifer forests is expected to have a greater impact on the water balance than the shift from agroforest to broadleaf (Fig. 9a and 9b). Compared to broadleaf forests, conifer forests exhibit higher simulated transpiration in March (Fig. 9b), driven by increased potential evapotranspiration and a relatively higher leaf area index (LAI) under wet soil conditions. This difference diminishes as the LAI of broadleaf forests increases in spring.



Changes in the LAI scaling factor influence water balance components in summer, increasing transpiration and
canopy evaporation while reducing recharge and soil evaporation (Fig. 9c and 9d). Increasing the LAI scaling
factor from 0.4 to 1.0 has a greater impact than reducing it from 1.6 to 1.0, as vegetation water use responds more
sensitively at low LAI values but plateaus at higher values due to energy or soil moisture limitations. Altering the
forest root parameter (β), while using the same LAI time series, primarily affects deep-layer transpiration,
reducing total transpiration and increasing recharge. Other water balance components remain unchanged because
the LAI time series is held constant.
Figure 10 illustrates the relative monthly deviations in evapotranspiration (ET) and groundwater recharge under
varying forest types, LAI scaling factors, and root distributions, relative to a baseline broadleaf forest.
Agroforestry increases recharge during the low-flow season (June–December) (Fig. 10a), while conifer forests
consistently reduce recharge and exhibit substantially higher ET in winter (Fig. 10b). The effects of LAI scaling
are most pronounced during the low-flow season. A higher LAI (scaling factor = 1.6) increases ET and reduces
recharge, whereas a lower LAI (scaling factor = 0.4) has the opposite effect. However, at higher LAI values, the
magnitude of relative deviation diminishes, suggesting a saturation effect. Root distribution also affects seasonal
water balance. Scenarios with deeper roots tend to reduce recharge, while shallow root systems enhance recharge
during dry months across all forest types. Overall, these results highlight the sensitivity of summer water balance
to vegetation structure. Agroforestry consistently exhibits more ecohydrologically resilient responses than conifer
forests, particularly under drought-sensitive conditions.




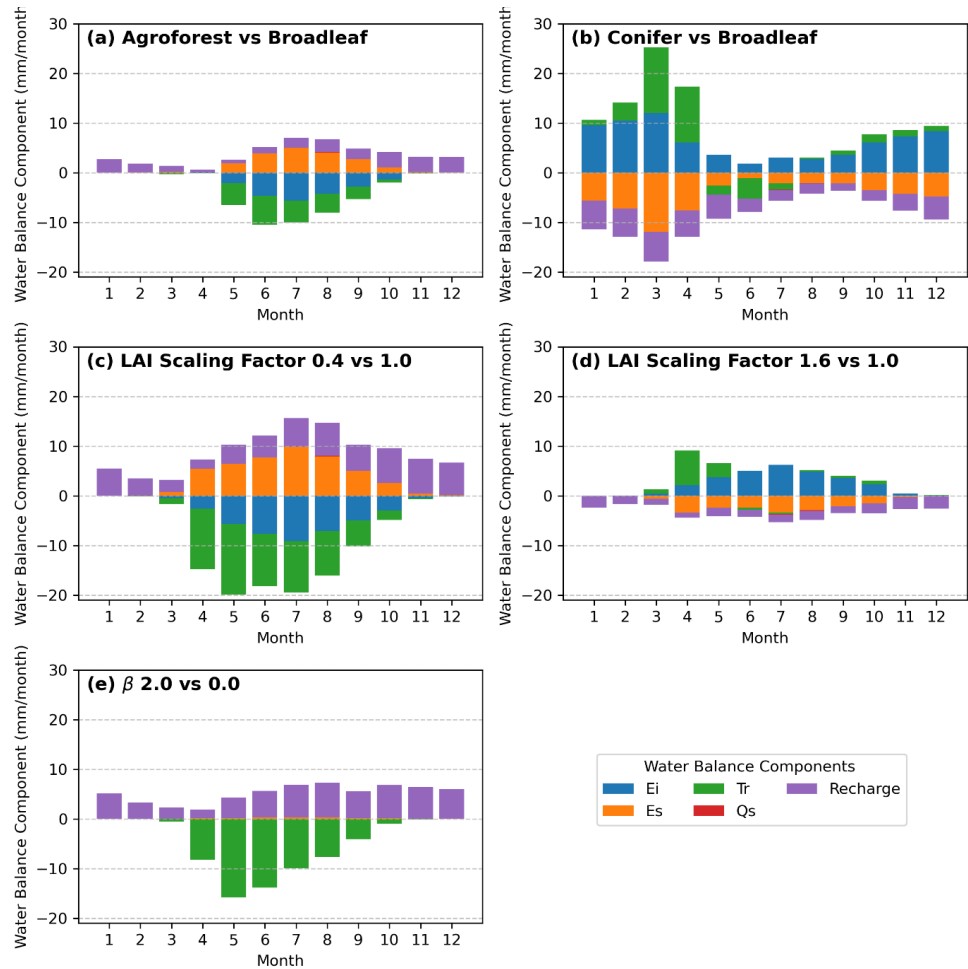

**Figure 9.** Monthly deviations of water balance components relative to the baseline broadleaf forest scenario. Each panel illustrates the deviation of monthly water balance components from the baseline simulation, with only one parameter modified in each scenario: (a) Agroforest, (b) Conifer forest, (c) LAI scaling factor = 0.4, (d) LAI scaling factor = 1.6, and (e) Root parameter $\beta$ = 2.0. Tr: transpiration, Ei: canopy evaporation, Es: soil evaporation, Qs: surface runoff, Recharge: groundwater recharge.

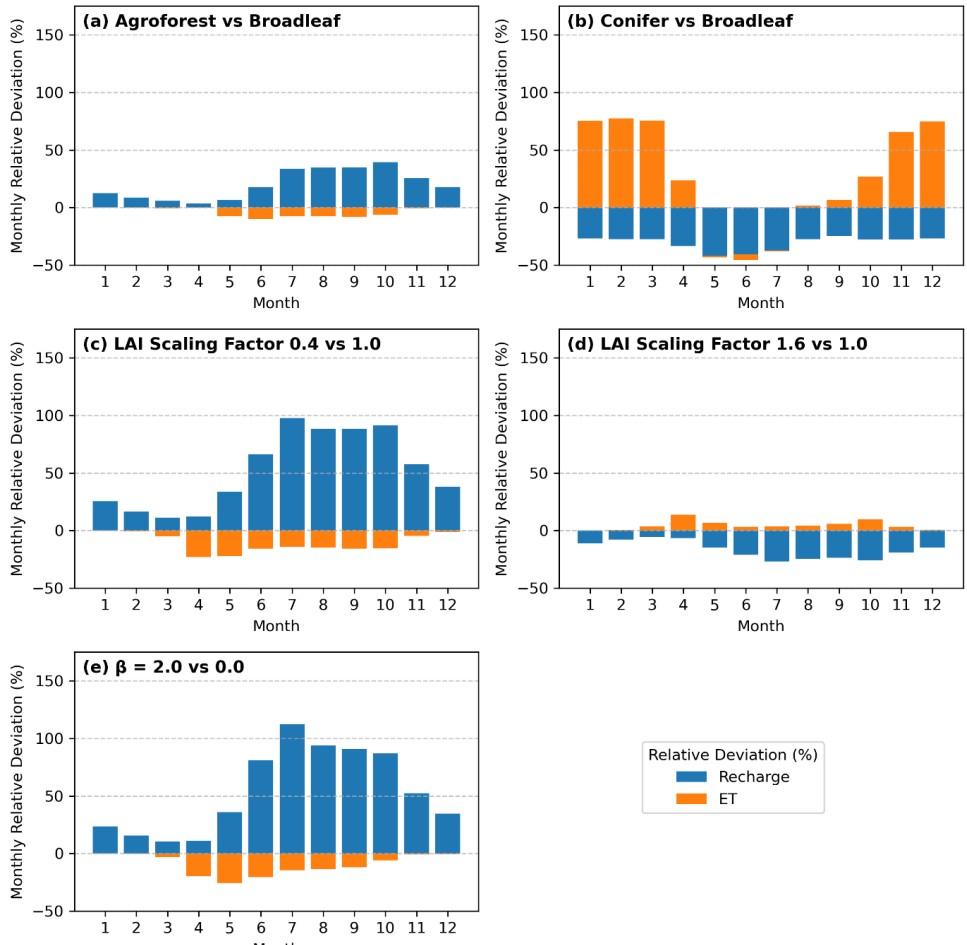

**Figure 10.** Monthly relative deviations in evapotranspiration (ET) and groundwater recharge, calculated as (scenario - basedline)/baslinline) ×100%, relative to the baseline broadleaf forest simulation. Each panel represents a different scenario in which one variable is modified while others are held constant: (a) Agroforest vs Broadleaf, (b) Conifer vs Broadleaf, (c) LAI scaling 0.4 vs 1.0, (d) LAI scaling 1.6 vs 1.0, and (e) Root parameter $\beta$ = 2.0 vs 0.0.

### 4.3.4 Soil Moisture Anomalies

Figure 11 shows the relative summer soil moisture anomalies across three forest types and three soil layers. Anomalies are calculated as the percentage deviation from the long-term seasonal mean, enabling normalized comparison across forest types and soil layers. Conifer forests exhibit the strongest soil moisture anomalies, followed by broadleaf forests, while agroforests exhibit the least variability, indicating greater stability in soil moisture. Furthermore, among the three soil layers, the intermediate layer (10–30 cm) consistently shows stronger anomalies across all forest types, with magnitudes nearly double those of the other layers, highlighting its vulnerability during summer drought. In contrast, the surface layer (0–10 cm) and deep layer (30–100 cm) exhibit weaker anomalies, likely due to frequent soil moisture replenishment by summer rainfall in the surface layer and either more stable moisture retention or greater water storage capacity at depth that compensates for drought





impacts. Negative soil moisture anomalies are more pronounced in summer than in spring, reflecting the stronger
seasonal drought effects and fluctuations in soil moisture (see Figure S5 in the Supplementary Material). During
spring, broadleaf forests and agroforests display similar negative soil moisture anomalies, suggesting comparable
seasonal soil moisture dynamics between these forest types (Figure S5).

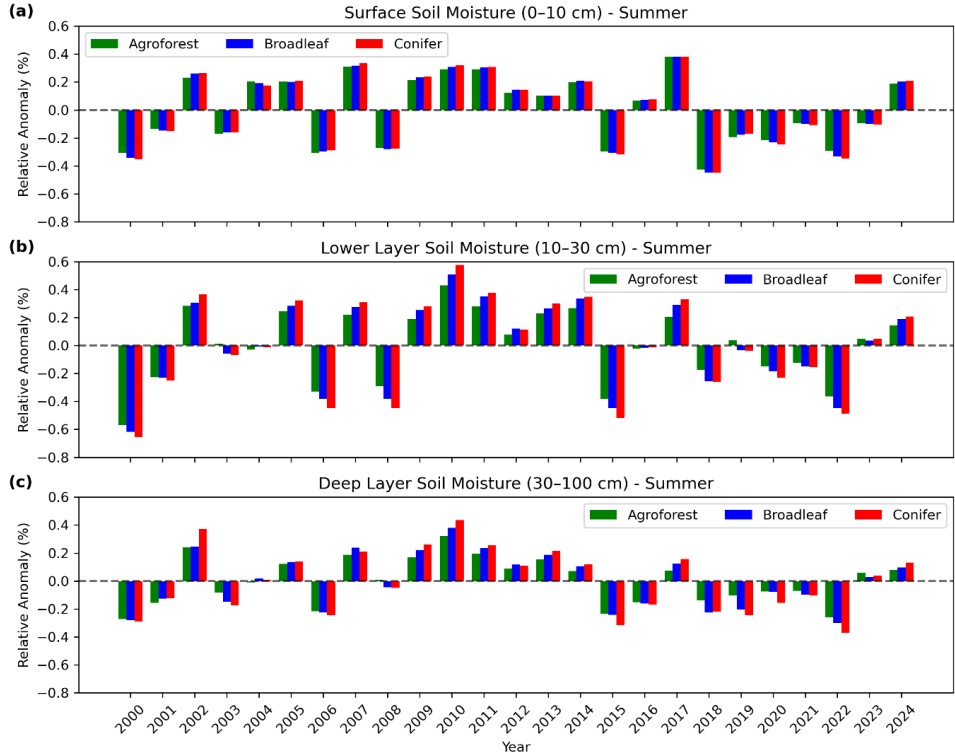

**Figure 11.** Relative soil moisture anomalies for summer (June–August) across three soil layers: (a) surface (0–10
cm), (b) lower layer (10–30 cm), and (c) deep layer (30–100 cm) for three forest types (Agroforest, Broadleaf,
Conifer). Bars represent deviations from the long-term mean, with positive values indicating wetter conditions
and negative values indicating drier conditions.
**5 Discussion**
**5.1 Implications of Forest Management Scenarios for Water Availability and Water Resource Management**
Assessing the influence of different land use types on water availability is inherently challenging due to the
complex interactions among vegetation, climate, and soil properties (te Wierik et al., 2021; Zhang et al., 2001).
Different vegetation types have distinct water demands, and their contrasting canopy structures affect how
precipitation is intercepted, and partitioned into infiltration, runoff, groundwater recharge, and evapotranspiration
(Brauman et al., 2010). Vegetation management practices can significantly alter these processes. Moreover, the
effects of vegetation and canopy structure may vary depending on underlying soil characteristic (Geris et al.,
2015). This complexity poses a significant challenge for land managers and policymakers, especially in drought-
sensitive regions facing increasing aridity due to climate change (Orth & Destouni, 2018). In such contexts,





providing informed guidance on sustainable land cover choices is increasingly important to maintain long-term
water availability (Estrela & Vargas, 2012). In regions where forestry has traditionally been an important land use,
shifting hydroclimatic conditions underscore the need to assess the resilience of different forest types and
management practices (Quandt et al., 2023). This requires evaluating water yield across multiple temporal scales,
including how forest management affects annual and seasonal water partitioning, and its implications for residual
water availability—specifically streamflow generation and groundwater recharge during low-flow periods
(Brown et al., 2005; Neill et al., 2021).
Although complex, process-based ecohydrological models such as RHESSys and EcH$_2$O can capture detailed
interactions among hydrological processes and water fluxes in data-rich research settings, their broader application
in forest and land management is often limited by the high data requirements for model forcing and calibration
(Fatichi et al., 2012; Kuppel et al., 2018; Tague & Band, 2004). In this study, we sought to apply a parsimonious
tracer-aided modelling approach to provide insights into the effects of different forest management scenarios on
water partitioning and land use resilience in Brandenburg, northeastern Germany, where recent droughts have
shown that traditional forest management practices focused on coniferous plantations of Scots pine may not be
sustainable (Luo et al., 2024). By employing the tracer-aided ecohydrological model EcoPlot-iso, we used a
generic approach to help quantify the long-term impacts of variations in forest type, stand density and root depth
distribution on both blue and green water fluxes.
In the baseline simulation for mature broadleaved forest, the estimated mean annual evapotranspiration (ET) for
or 2000–2024 was 390 mm/year, accounting for 72% of annual precipitation. This value is consistent with ET
estimates reported in previous modelling at DMC in 2021, ranging from 68% to over 80% of annual precipitation
(Landgraf et al., 2023), 2018–2020 (Smith et al., 2021). The discrepancy may reflect interannual climate
variability and the influence of particularly dry or wet years that cannot be captured by short-term assessments.
Differences in model structure, parameterization, and input data may also contribute to the spread in ET values.
Nonetheless, this comparison underscores the importance of long-term simulations for capturing representative
hydrological behavior and evaluating the impacts of forest management strategies under variable climatic
conditions.
In catchments like DMC, where evapotranspiration (ET) is high, atmospheric demand is the primary driver of root
water uptake, though vegetation plays a key role in regulating its impact on water availability. In Brandenburg,
coniferous forests have traditionally been favored on sandy soils, but modelling indicates high water use due to
interception losses and year-round transpiration potential (Fig. 9). Consequently, the implications for both reduced
groundwater recharge and reduced forest productivity has encouraged landowners to explore alternative land use,
such as broadleaves forests and agroforestry. These options have the potential for optimizing biomass productivity
and land use resilience with increased landscape water retention and increased groundwater recharge.
These results (e.g., Figs. 7 and 8) have practical applications, such as estimating the direction and magnitude of
the changes in evapotranspiration and water yield as a function of forest management practices, driven by
alterations in canopy structure and rooting depth. The modelling approach thus provides useful insights into the
hydrological implications of alternative canopy structures and rooting patterns for water use. Figure 12 compares
the mean annual partitioning of water fluxes and soil moisture across broadleaf, coniferous, and agroforest types
under dry and wet year conditions. It highlights how different vegetation strategies influence hydrological



resilience, with substantial differences in water partitioning observed between dry and wet years across contrasting
forest management scenarios. By simulating long-term water availability across periods of alternating wet and dry
conditions, EcoPlot-iso simulations suggest that mixed forests and agroforestry can enhance water supply
resilience in drought-sensitive catchments by sustaining both water yield and groundwater recharge.





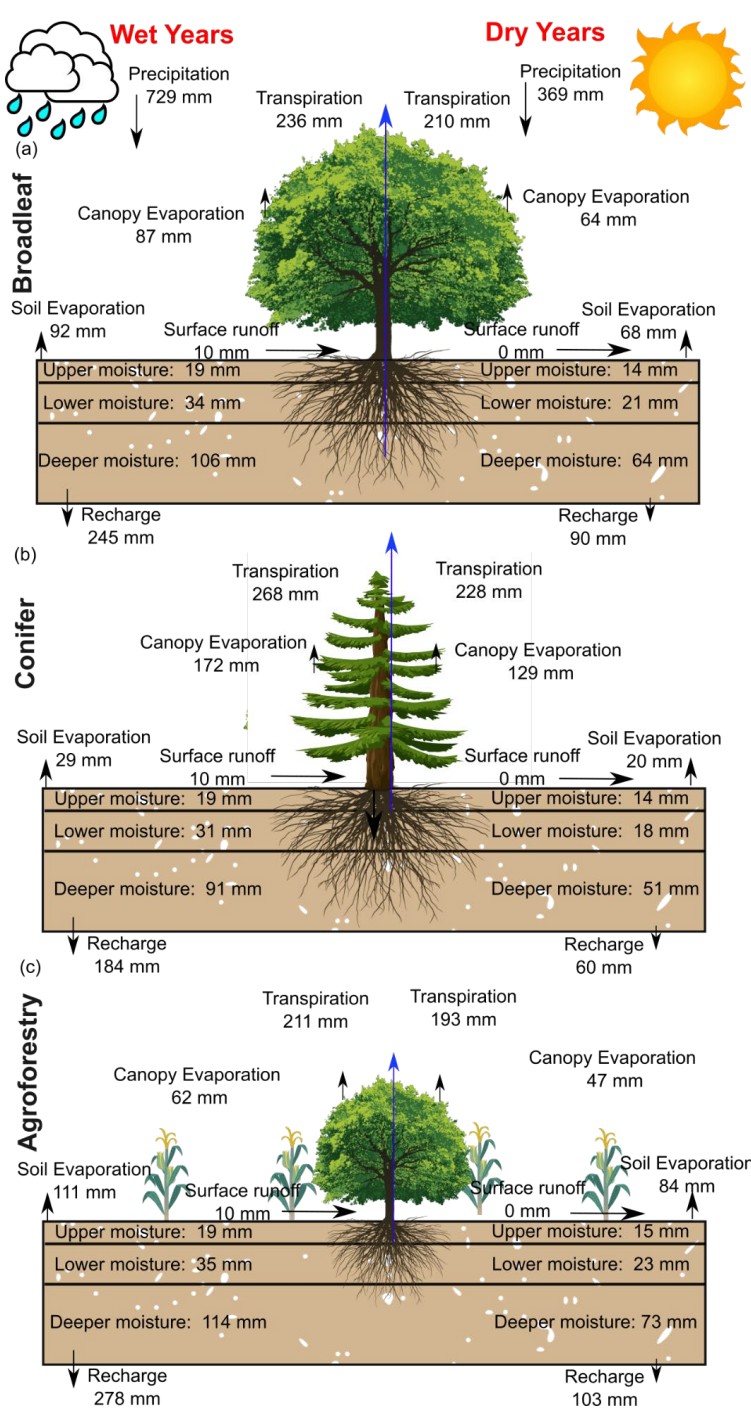

**Figure 12.** Comparison of mean annual water fluxes and soil moisture in the upper, lower, and deeper layers for Broadleaf (a), Coniferous (b), and Agroforest (c) forests under dry (2006, 2018, 2022) and wet (2002, 2007, 2010, 2023) year conditions.



**5.2 Soil Moisture Anomalies and Their Implications for Resilience**

At most of the monitoring plots in the DMC, groundwater is typically more than 3 meters below the ground surface (Ying et al., 2025). Therefore, except in older forest plots with deeply rooting trees, vegetation relies on soil moisture for root water uptake. Even for mature trees, there is evidence that most root water uptake occurs in the near-surface soil horizons, as demonstrated by (Birkel et al., 2025), 20 km from the DMC. A global synthesis by Evaristo & McDonnell (2017) further supports this, indicating that ~77% of plant water uptake comes from shallow sources, with deeper groundwater use primarily in more arid regions. While hydraulic redistribution may provide deeper access for some species (Emerman & Dawson, 1996), rooting strategies are complex and highly species-specific (Demir et al., 2024). In this context, our results highlight the intermediate soil layer (10–30 cm) as the most reactive and significant for sustaining transpiration, with anomaly magnitudes nearly twice those of both the shallow (0–10 cm) and deeper (30–60 cm) layers across all forest types.

In addition, seasonal comparisons revealed that summer soil moisture anomalies were more negative than those in spring for all forest types (Figure S5). This is likely linked to higher temperatures and evapotranspiration during summer, which intensify water stress and drive seasonal variation in soil moisture availability. Forest density and rooting characteristics substantially influenced the relative magnitude of soil moisture anomalies (Figure S6 and S7, respectively). Denser forests exhibited stronger negative anomalies during dry periods and enhanced positive anomalies in wet periods, amplifying seasonal fluctuations. For example, high-density (LAI scaling factor 1.6) conifer stands showed relative anomalies up to 25% greater than their low-density counterparts (Figure S7). In contrast, shallow-rooted systems moderated this response, leading to more stable soil moisture dynamics. Among the management scenarios, agroforestry consistently exhibited the smallest anomalies, reflecting greater buffering capacity and higher ecohydrological resilience.

The improved rooting scheme in EcoPlot-iso represents depth-dependent transpiration by dynamically linking root water uptake efficiency to soil moisture availability across three soil compartments (see Section 3.2). Unlike models such as RHESSys and EcH2O, which partition a prescribed total transpiration—typically derived from the energy balance—across layers based on static root distributions, our approach allows transpiration to emerge from potential evapotranspiration, root-zone constraints, and soil moisture availability. The aim was not to optimize species-specific root dynamics, but to represent the relative influence of rooting depth on water uptake and partitioning, particularly in shallow-rooted or structurally diverse systems such as young forests. While the new implementation improves the process representation of root–soil interactions, it did not result in a substantial improvement in simulated soil moisture. For shallow vegetation types such as grasslands and croplands, model performance—measured using the mKGE was similar with and without the new transpiration function (results not shown). Moreover, direct validation of the root uptake scheme remains challenging due to the lack of supporting observations, such as root distribution data, xylem water isotopes, or sap flux measurements. Addressing this issue is a clear priority for future research.

These findings highlight how structurally diverse systems, such as agroforests, enhance the buffering capacity of ecosystems by improving groundwater recharge and reducing the amplitude of soil moisture fluctuations, thereby supporting greater resilience during dry periods (Tetzlaff et al., 2024). Together, these insights underscore the importance of rooting depth, forest structure, and seasonal climate variability in shaping soil moisture patterns



and regulating vegetation resilience. Accounting for these factors is essential for informing adaptive forest
management in drought-prone catchments like the DMC.

### 5.3 Advancing Tracer-Aided Ecohydrological Modeling: Challenges and Future Outlook

This study demonstrates that tracer-aided ecohydrological models, such as the isotope-aided EcoPlot-iso, can
effectively quantify the impact of forest management scenarios on water partitioning and ecohydrological
resilience. By distinguishing between evaporation, transpiration, and subsurface water movements using stable
isotopes (Tetzlaff & Soulsby, 2008), the model captures key hydrological responses—including
evapotranspiration (ET), groundwater recharge, and soil moisture dynamics—under varying management
strategies. These insights support evidence-based decision-making in drought-sensitive landscapes.
Despite these advances, several challenges remain. Conducted in a 66 km² mid-sized basin, this study did not
include land use change induced atmospheric feedbacks—such as changes in albedo, radiative balance, or rainfall
patterns—which are less critical at this scale but become important in larger-scale modeling (Ellison et al., 2012;
Filoso et al., 2017). Moreover, this study applied a multi-objective calibration approach, combined with Monte
Carlo sampling, that equally weighted isotopic and soil moisture data. However, further investigation is needed
in how these observational constraints are balanced and interpreted. Recent advances—such as the
DREAM(LoAX) framework (Wu et al., 2025)—demonstrate how simultaneous calibration and diagnostic
analysis under the equifinality thesis can improve parameter identifiability, model robustness, and process
understanding in tracer-aided ecohydrological models.
Many recent studies have used isotopic data to investigate root water uptake patterns, revealing how tree species,
soil properties, and spatial water availability shape plant water use strategies (Demir et al., 2024; Rothfuss &
Javaux, 2017). Integrating tracer-aided models with soil and xylem water isotope data offers a promising path to
improving the representation of root water uptake, which is often simplified in current modelling approaches
(Birkel et al., 2025). Improving root uptake representation requires consideration of species-specific traits and
local soil-water conditions. However, the practical application of such improvements is limited by the scarcity of
soil and xylem water isotope data, which are essential for constraining root water uptake dynamics but remain
rare due to the labor-intensive and technically demanding nature of field sampling and laboratory analysis
(Landgraf et al., 2022; Sprenger et al., 2017). This scarcity hinders the spatial and temporal resolution of
observational data, limiting our ability to refine root water uptake processes in tracer-aided models.
Upscaling from plot to landscape level remains complex due to spatial heterogeneity in vegetation, soils, and
topography. Addressing this requires spatially distributed modeling frameworks that can explicitly capture
heterogeneity in ecohydrological processes across different landscape units (Kuppel et al., 2018; van Huijgevoort
et al., 2016). Enhanced integration with remote sensing techniques can also help address these scaling limitations
by providing spatially continuous data on vegetation dynamics, soil moisture, and ET (Yang et al., 2023).
Incorporating ET observations, for instance, could strengthen model interpretation of flux dynamics. Currently,
key processes such as lateral subsurface flows and upward capillary fluxes are not explicitly represented in the
EcoPlot-iso model. Including these components, along with improved representation of groundwater-surface
water interactions, could improve simulations of water connectivity and storage resilience.





Future development should emphasize the coupling of tracer-based approaches with high-resolution hydrological
modeling, remote sensing data, isotope data, and empirical field studies. Such interdisciplinary integration is
essential for improving the scalability and applicability of tracer-aided ecohydrological models, especially for
informing sustainable forest and water management under uncertain hydroclimatic futures.
**6 Conclusion and Outlook**
The isotope-aided EcoPlot-iso modelling framework was applied to quantify the impacts of different forest
management strategies on water partitioning and ecohydrological resilience in the drought-sensitive Demnitzer
Millcreek catchment in northeastern Germany. The model was first set up and evaluated under a baseline
simulation for the period 2000–2024 at a broadleaf reference site, successfully reproducing observed soil moisture
and soil water isotope dynamics using a multi-objective calibration approach. A novel depth-dependent root water
uptake function was integrated, and a suite of scenario simulations—varying in forest type, canopy density, and
rooting depth—was conducted to assess changes in evapotranspiration, groundwater recharge, and soil moisture
anomalies under both dry and wet climatic conditions.
The results revealed clear trade-offs between evapotranspiration (ET) and groundwater recharge, depending on
forest management strategies. Coniferous forests intensified drought impacts, with approximately 8–13% higher
ET compared to broadleaf and agroforestry systems, and significantly reduced groundwater recharge, particularly
during low-flow dry periods. In contrast, agroforestry systems effectively buffered drought stress and maintaining
lower soil moisture variability, which simultaneously lowering ET and enhancing groundwater recharge by about
13%. Further analysis highlighted contrasting ecosystem responses: conifers showed the strongest soil moisture
anomalies, indicating greater drought sensitivity, while agroforests exhibited the most stable soil water storage.
The intermediate soil layer (10–30 cm) was identified as the most responsive zone, consistently exhibiting the
largest anomalies due to its role as the dominant root water uptake region supporting transpiration.
Beyond advancing process understanding, this study provided practical tools for land management. By
incorporating key controls such as canopy properties and root distribution, EcoPlot-iso facilitates an accessible
means of assessing long-term land management impacts on landscape ecohydrology. The visualization and
decision-support framework developed here offers a transparent, scenario-based platform for evaluating forest
management strategies in climate-sensitive regions. These tools are well-suited for informing resilient land use
planning under increasing climate variability.
Looking ahead, future research could usefully aim to incorporate additional isotopic tracers—such as deeper soil
water (> 1 m), groundwater, and xylem water isotopes—to further constrain root water uptake functions and
capture their variability across species and hydroclimatic conditions. The integration of high-resolution remote
sensing data—particularly LiDAR for detailed characterization of forest structure—will enhance model
parameterization and improve the spatial representation of heterogeneity in canopy height, leaf area distribution,
and forest density. Advancing the EcoPlot-iso framework to incorporate lateral subsurface flows, groundwater
dynamics, and coupled land–atmosphere feedbacks will support broader applications, including the assessment of
large-scale land use change. Collectively, these developments will enhance model robustness and enable more
informed, resilient land and water management strategies under a warming climate.



**Code and data availability**

The data and code that support the findings of this study are available from the corresponding author upon reasonable request.

**Author contribution**

CJ contributed to the methodology, software development, formal analysis, investigation, visualization, and writing of the original draft. DT contributed to conceptualization, investigation, data curation, validation, resources, project administration, and funding acquisition. SW contributed to methodology, investigation and data curation. CB contributed to software, methodology, and resources. HL contributed to investigation, visualization and validation. CS contributed to conceptualization, methodology, validation, investigation. All authors contributed to writing – review and editing.

**Competing interests**

The authors declare that they have no conflict of interest.

**Acknowledgements**

Tetzlaff's contributions were partly funded through the WETSCAPES2.0 project (DFG TRR410/1 2025). Tetzlaff also received funding from the "Wasserressourcenpreis 2024" awarded by the Rüdiger Kurt Bode-Foundation. Contributions from Soulsby were supported by Leibnitz Association Germany in the project Wetland Restoration in Peatlands. Laudon was funded by KAW 2018.0259 and 2023.0245, and Soulsby was also funded as an International KSLA Guest Professor at SLU by the Wallenberg Foundation (WP2023-0001). Birkel would like to thank the IGB for generously supporting him with a senior fellowship and the UCR for a sabbatical license. We extend our appreciation to Benedikt Boesel and the team from the Finck Foundation (www.finck-stiftung.org) for their collaborative support and for granting access to study sites.

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
