# Peer review of "Assessing the drought resilience of different land management scenarios using a tracer-aided"

_EGUsphere, 2025_

## Author Comment (AC1)

**Response to Referee Comment #1:**

We thank the referee for the careful review and insightful suggestions. We believe that addressing the comments will strengthen the paper and improve the clarity of our key findings. From the reviewer's comments, we can see that we did not explain some aspects sufficiently well. Below, we provide a point-by-point response to each comment, indicating the changes that we will make in the revised manuscript to address the referee's questions and suggestions.

Sincerely,

Dr. Cong Jiang (on behalf of all co-authors)

**Major Comments:**

1) The study differentiates forest types by adjustments to LAI. All other aspects of forest functioning (i.e., rooting depths, sensitivities to low soil moisture anomalies, water needs, etc.) are presumed to be identical. My intuition tells me that these other factors do matter for accurate simulation of the hydrological fluxes this paper seeks to explore. The sensitivity tests demonstrate to me that uncertainty attributable to rooting depths alone greatly overwhelms the differences estimate across forest types. I understand the desire for a simple modeling framework to estimate the benefits of forest management, but this seems maybe too simplistic. Is there a way to better represent the functional differences across the forest types beyond canopy LAI?

Reply to Q1:

We thank the reviewer for summarizing our approach and highlighting her/his concerns so concisely.

Of course, forests differ in many functional traits beyond canopy LAI which affect water partitioning; including those that we also consider (e.g. rooting depth) and others which we do not (stomatal conductance, VPD sensitivity, and tolerance to soil water stress). In this study, however, our intention was to adopt a parsimonious and generic framework to assess the dominant effects on water partitioning in our study area through a modelling experiment, rather than to reproduce detailed species-specific physiology. In our revision, we will make this exploratory nature of the study clearer. Our tracer-aided framework is new, though based on the proven and widely-applied EcoPlot model. In this application we sought to systematically explore variation in three key dimensions of forest characteristics that strongly influence water fluxes in out study area: (i) canopy density (via LAI scaling factors), (ii) rooting distribution (via the $\beta$ parameter), and (iii) generic forest type (broadleaf, conifer, agroforest, each represented by distinct MODIS-derived and bias-corrected LAI time series).

In the revision, we will acknowledge that this simplified representation does not capture all species-level effects. However, the modelling experiment was deliberately designed to test the relative influence of canopy and rooting traits under comparable conditions, and to provide scenario-based insights into long-term water partitioning under alternative management strategies in a data-limited setting. We will strengthen the Discussion to acknowledge this limitation and to note that incorporating additional functional traits (e.g., stomatal conductance, VPD sensitivity) may be critical at other sites and is therefore an important avenue for future model development. We will also adjust the title of the paper to reflect its more exploratory nature.

2) The LAI scaling that is performed is the only major factor that differentiates the forest types. I was unclear on exactly how this scaling was performed and justified. Given that this is the only primary difference considered I would expect to see a strong justification of the approach.

Reply to Q2:

We thank the reviewer for raising this point. To clarify, the differentiation of forest types in our study was not only based on LAI scaling, but also on the use of forest-type–specific LAI time series derived from the MODIS product (2000–2024), which capture realistic seasonal and interannual variability. The LAI scaling was then applied to these type-specific trajectories to represent changes in stand density within a given forest type (e.g., thinning or regeneration), rather than to differentiate forest types per se. Thus, forest type is represented by distinct observed LAI trajectories (broadleaf, conifer, agroforest), while the scaling factors capture management-induced density changes. We would also emphasize that the LAI changes should also be interpreted in the context of the different rooting depths. We will ensure that this is clearer in the revision.

A similar LAI scaling factor method has been used in tracer-aided modelling by the author team before; for example, Neill et al. (2021) applied scaling factors of 0.04–1.37 with EcH2O-iso to examine the effects of natural forest regeneration on water flux partitioning, water ages, and hydrological connectivity. In our case, the applied scaling range (0.2–1.8) is deliberately broader, allowing us to explore canopy density scenarios from strongly thinned to very dense stands. The potential maximum LAI of forests is difficult to define, as it depends not only on stand age, tree density, and site conditions, but also on broader environmental gradients such as soil fertility, acidity, and precipitation. Nevertheless, values up to 9.5 m² m⁻² have been reported for mature Central German beech forests (Leuschner et al., 2006). Thus, although our scaling range is intentionally broad, the resulting scaled LAI values remain within the variability observed in European forests and provide a plausible envelope for management-induced changes in canopy density.

We will expand Section 3.3 to explicitly explain the derivation and justification of this scaling range, and to emphasize that the scaled LAI values remain within realistic ranges for each forest type and also include this in the discussion in the revision

Reference:

Neill, A. J., Birkel, C., Maneta, M. P., Tetzlaff, D., & Soulsby, C. (2021). Structural changes to forests during regeneration affect water flux partitioning, water ages and hydrological connectivity: Insights from tracer-aided ecohydrological modelling. Hydrology and Earth System Sciences, 25(9), 4861-4886. https://doi.org/10.5194/hess-25-4861-2021

Leuschner, C., Voß, S., Foetzki, A., & Clases, Y. (2006). Variation in leaf area index and stand leaf mass of European beech across gradients of soil acidity and precipitation. Plant Ecology, 186(2), 247-258. https://doi.org/10.1007/s11258-006-9127-2

**Specific Comments:**

Q3. Line 45: The use of "resilience" here is a little unclear to me. Are you talking about resilience of the vegetation or resilience of human-accessible water?

Reply to Q3:

Thank you for point this out. In our study, we use "ecohydrological resilience to drought" in an ecohydrological sense (as defined in Tetzlaff et al., 2024), referring to the ability of the soil–plant–water system to sustain green-water use (evapotranspiration) while maintaining blue-water availability (runoff, recharge) under drought stress. This definition goes beyond vegetation or human-accessible water alone by considering their coupled dynamics. We will clarify this in the Introduction in the revision.

The revised text will read:

"Consequently, forest management decisions, (e.g., afforestation, thinning, species selection etc.) can significantly affect water yield, the partitioning into blue (runoff, groundwater recharge) and green (evapotranspiration) water fluxes, and the ecohydrological resilience to drought (Tetzlaff et al., 2024) — the ability of soil–plant–water system to sustain key hydrological and ecological functions under stress (Falkenmark & Rockström, 2006; Neill et al., 2021))."

Q4. Line 51 – 52: The central hypothesis of this work seems to be "does land management use and forest management impact water fluxes?" Here you are stating that this is the case. Consider rewording.

Reply to Q4:

We thank the reviewer for the detailed observation. We agree the current phrasing may read as a statement of fact, and we will reword it to highlight research motivation instead of a conclusion.

The revised text will read:

"Because land use practices—particularly forest management—are expected to strongly influence water partitioning, it is important to assess their impacts under changing hydroclimatic conditions in order to evaluate the ecohydrological resilience of water and land systems, especially in drought-prone areas."

Q5. Line 65: To my knowledge, EcH2O-iso does include species defined root water uptake functions. This sentence makes it sound as if that is not the case. Please edit if I am right about this.

Reply to Q5:

We thank the reviewer for pointing this out.

You are correct that EcH2O-iso includes species-defined root water uptake functions, where total transpiration is first calculated from the energy balance and then partitioned across soil layers based on soil water content and an exponential root distribution parameter (Kroot). Our sentence was imprecise and we will revise it to clarify that, while EcH2O-iso (and other models such as mHM) incorporate root water uptake functions, the dynamic and plastic nature of root system development (e.g. interannual variability in maximum rooting depth under different management or climate conditions) remains inadequately represented.

In addition, our model EcoPlot-iso differs in structure: because it does not include an energy-balance module, total transpiration is not calculated a priori. Instead, we use an alternative "bucket-style" parameterisation, where transpiration is sequentially satisfied from upper to deeper soil layers according to the relative soil moisture availability (STO/Smax) of each layer, while evaporation is governed by canopy fraction and topsoil water content (Eqs. S5–S9).

The revised text will read:

"Although many ecohydrological models include parameterised root water uptake functions (e.g., mHM, EcH2O-iso), the dynamic and plastic nature of root distribution and maximum rooting depth is usually simplified. In contrast, EcoPlot-iso applies a bucket-style approach without an explicit energy balance, and in this study, we extend the model with a new parameterisation to explicitly account for rooting depth variability across contrasting forest management scenarios."

Q6. Lines 62-93: This all feels more like discussion to me. The central aim of the paper is about how forest management impacts water cycling. These paragraphs are a good discussion of model complexity that are relevant but tangential to the central aim. I would recommend that the authors consider reorganizing.

Reply to Q6: We thank the reviewer for this constructive suggestion. The current text in Lines 62–93 was intended to introduce the challenges of modelling ecohydrological processes and forest-management scenarios and to motivate our use of a tracer-aided, intermediate-complexity model. We agree that the present level of technical detail is better placed in the Discussion. We will shorten Lines 62–93 to 3-4 sentences so the Introduction remains tightly focused on the study aim (how forest management affects water cycling), and we will move the detailed material on model complexity, root-uptake function, and tracer-based methods into the Discussion to compare alternative modelling approaches and to highlight limitations and remaining research gaps.

Q7. Line 172: Are these depth variations selected by the user? How do we know that they are appropriate to capture vertical variations in soil isotopic compositions and root water uptake?

Reply to Q7:
Thank you for this helpful comment. The vertical discretization into three soil layers (0–10 cm, 10–30 cm, and 30–100 cm) follows the standard configuration of EcoPlot-iso, which has been successfully applied in several recent studies in this region (e.g., Landgraf et al., 2023). Multi-layer soil structures of this type are also common in widely applied hydrological and land surface models. For example, Noah-MP typically employs a four-layer scheme (0–10, 10–40, 40–100, and 100–200 cm), and CLM also uses a comparable multilayer configuration, although the exact depth intervals differ.

Moreover, in this study soil water isotopes were sampled at finer vertical resolution (0–5, 5–10, 10–20, 20–30, and 30–50 cm; line 233). Soil moisture was measured with handheld probes at 0–10 cm and with permanently installed sensors at 15–20 cm and 40–100 cm (Table S1). These depths and observations align with the three model layers (0–10, 10–30, 30–100 cm), providing higher-resolution observations within each layer to support calibration and validation of soil moisture and soil water isotopic dynamics. We will double check that this is clarified in the revision.

Overall, such discretizations have proven effective for representing vertical variations in soil moisture, and isotopic compositions. Although the discretization can be adjusted in principle, we adopted this configuration to ensure comparability with previous applications and consistency with established practice.

Reference:
Landgraf, J., Tetzlaff, D., Birkel, C., Stevenson, J. L., & Soulsby, C. (2023). Assessing land use effects on ecohydrological partitioning in the critical zone through isotope-aided modelling. Earth Surface Processes and Landforms, 48(15), 3199–3219. https://doi.org/10.1002/esp.5691

Niu, G. Y., Yang, Z. L., Mitchell, K. E., Chen, F., Ek, M. B., Barlage, M., ... & Xia, Y. (2011). The community Noah land surface model with multiparameterization options (Noah‑MP): 1. Model description and evaluation with local‑scale measurements. Journal of Geophysical Research: Atmospheres, 116(D12). https://doi.org/10.1029/2010JD015139

Q8. Section 3.2: Do the different vegetation classes differ in any ways other that the beta term defining the decay of root water uptake demand?
Reply to Q8:
Yes — each vegetation class / site is calibrated separately. Calibration includes not only the $\beta$ decay parameter but also additional vegetation and soil parameters (Table S3). We will clarify that vegetation classes differ by these broader sets of parameters, not only by the $\beta$ root-decay term.
In our study, EcoPlot-iso was applied to four contrasting land-use sites: broadleaf forest, cropland, agroforestry, and grassland. The broadleaf forest site was used as the baseline; from its calibration we retained the 100 best-performing simulations and corresponding parameter sets. As we know, this site represents a mature forest, for which $\beta = 0$ was set, implying no constraints on transpiration from root distribution within the 0–100 cm soil profile.
For the forest management scenarios, we then varied three key dimensions of forest characteristics — generic forest type, forest density, and root distribution — considered separately. This allowed us to test how differences in other vegetation canopy and root distribution and influence water uptake and ecohydrological resilience.

Q9. Line 193: I'm a little unclear from this description if the model simulates multiple vegetation classes within the same simulation. This sounds like only one is possible. How then does this support study of intercropping that you discuss previously?
Reply to Q9:
EcoPlot-iso is applied at the plot scale and typically represents a single dominant vegetation type per simulation, rather than explicitly simulating multiple vegetation classes within the same run. For agroforestry or intercropping, this is approximated by applying MODIS-derived LAI and calibrating canopy and soil parameters to represent mixed crop–tree systems, rather than simulating multiple vegetation types simultaneously. We will clarify this point and the agroforestry simulations more clearly in the revised manuscript (Methods section 3.3).

Q10. Line 218: The model computes E/T from PET? I'm guessing it uses some downregulation of PET to AET based on soil moisture in the root zone? Please specify the approach. Is there any consideration for how different canopy structures impact E/T? This was mentioned previously as an important consideration for understanding how forest management strategies impact water cycling and resilience.
Reply to Q10:
We will clarify in the Methods how PET is downregulated to AET and how canopy structure influences the E/T partitioning.
Specifically, a new paragraph and a summary table with variable definitions and governing equations will be added in the Supplementary Material (Table S2; Eqs. S1–S12), describing how interception, evaporation, transpiration, and soil evaporation are constrained by canopy storage and soil-moisture conditions. In brief, PET is partitioned into canopy and soil fractions based on canopy surface cover fraction SCF derived from LAI via Beer's law, and differences in canopy structure among management scenarios (broadleaf, conifer, agroforestry) are represented through their LAI time series (thus SCF), interception capacity, and rooting-depth distributions, which together control the E/T partitioning.
Text to be added in the Methods:
"In EcoPlot-iso, canopy surface cover fraction (SCF) is derived from LAI using Beer's law with an extinction coefficient $rE$ (Eq. S1). Maximum canopy storage is determined by SCF and an interception threshold parameter $\alpha$. Interception is represented by a nonlinear saturation-type function (Eq. S2), whereby precipitation is first stored in the canopy until maximum canopy storage is reached, and any excess is routed as throughfall. Potential evapotranspiration PET is partitioned into canopy and soil fractions according to SCF (Eqs. S3 and S4). The canopy fraction drives evaporation from the interception store and transpiration from the soil layers, while the soil fraction drives evaporation from

the upper soil layer (Eqs. S5–S9). Actual fluxes are constrained by water availability: interception evaporation depends on canopy storage, transpiration is sequentially satisfied from the upper to deeper soil layers according to the relative soil-moisture availability (STO/Smax) of each layer, and soil evaporation is limited by moisture availability in the upper soil. Surface runoff is represented using a Hortonian threshold approach, where precipitation in excess of infiltration capacity is routed as runoff (Eq. S10). Preferential flow is triggered when throughfall exceeds a threshold, with a calibrated parameter controlling the bypass proportion (Eq. S11). Percolation, compartment flow and groundwater recharge are represented as storage–discharge relationships, where outflows are parameterised as power functions of soil or groundwater storage (Eqs. S12-S14). Full variable definitions and governing equations are provided in the Supplementary Material (Eqs. S1–S14)."

**Table S2.** Ecohydrological processes in EcoPlot-iso: variable definitions and governing equations

| Variable | Description | Equation |
|---|---|---|
| $SCF$ | Surface cover fraction | $SCF = 1 - e^{(rE*LAI)}$  (1) |
| $Int$ | Canopy interception | $Int = (\alpha * \text{LAI}) * \left( 1 - \dfrac{1}{1 + \dfrac{SCF * P}{\alpha * LAI}} \right)$  (2) |
| $T_p$ | Canopy fraction of PET | $T_p = PET * SCF$  (3) |
| $E_p$ | Soil fraction of PET | $E_p = PET * (1 - SCF)$  (4) |
| $Ei$ | Canopy evaporation | $E_i = \begin{cases} Int_S & \text{if } T_p > Int_S \\ T_p & \text{if } T_p \leq Int_S \end{cases}$  (5) |
| $Es$ | Soil evaporation of upper soil layer | $E_s = E_p * \left( \dfrac{STO}{S_{max}} \right)$  (6) |
| $Tr\_Upper$ | Transpiration from the upper soil layer | $T_{p1} = (T_P - E_i) * \left( \dfrac{STO}{S_{max}} \right)$  (7) |
| $Tr\_Lower$ | Transpiration from the lower soil layer | $T_{p2} = (T_P - E_i - T_{p1}) * \left( \dfrac{GW}{GW_{max}} \right)$  (8) |
| $Tr\_Deep$ | Transpiration from the deep soil layer | $T_{p3} = (T_P - E_i - T_{p1} - T_{p2}) * \left( \dfrac{SDeep}{SDeep_{max}} \right)$  (9) |
| $Qs$ | Surface runoff | $Q_s = PN - I_c$  (10) |
| $Pref\_Flow$ | Preferential flow | $\text{Pref\_Flow} = PN * PF_{scale}$  (11) |

| | | |
|---|---|---|
| *Perc* | Percolation flux | $Perc = ks1 * \left( \dfrac{STO}{S_{max}} \right)^{g1}$   (12) |
| *Sdeep* | Compartment flow | $Sdeep = ks2 * \left( \dfrac{GW}{GW_{max}} \right)^{g2}$   (13) |
| *Recharge* | Groundwater recharge | $Recharge = ks3 * \left( \dfrac{SDeep}{SDeep_{max}} \right)^{g3}$   (14) |

Q11. Section 3.2 (line 189-199): Is there a reference for this root depth model? The equations look very similar to the water uptake equations of the SWAT model.

Reply to Q11:

The exponential root depth function that we implemented in EcoPlot-iso builds on earlier root uptake formulations, where uptake was assumed to decline with depth either linearly (Hoogland et al., 1981; Prasad, 1988) or exponentially (Li et al., 1999; Wu et al. 1999). Our implementation follows this exponential form and is conceptually similar to those used in models such as EcH2O-iso and SWAT. In EcoPlot-iso, the $\beta$ term serves as a fitted extinction coefficient and is the only parameter estimated in the root uptake function, representing the exponential decline of root uptake efficiency with depth. We will revise Section 3.2 to include the relevant references and clarify this.

Reference:

Hoogland, J. C., Feddes, R. A., & Belmans, C. (1981, March). Root water uptake model depending on soil water pressure head and maximum extraction rate. In III International Symposium on Water supply and Irrigation in the open and under Protected Cultivation 119 (pp. 123-136). https://doi.org/10.17660/ActaHortic.1981.119.11

Prasad, R. (1988). A linear root water uptake model. Journal of Hydrology, 99(3-4), 297-306. https://doi.org/10.1016/0022-1694(88)90055-8.

K. Y. Li, J. B. Boisvert, and R. De Jong. 1999. An exponential root-water-uptake model. Canadian Journal of Soil Science. 79(2): 333-343. https://doi.org/10.4141/S98-032

Wu, J., Zhang, R. & Gui, S. Modeling soil water movement with water uptake by roots. Plant and Soil 215, 7–17 (1999). https://doi.org/10.1023/A:1004702807951

Q12. Line 220: Linear interpolation?

Reply to Q12: Yes, we will incorporate this in the revised manuscript.

The revised text will read:

"The Leaf Area Index (LAI) was obtained from the MODIS 8-day resolution dataset and linearly interpolated to daily timesteps."

Q13. Line 226-229: Do you know the make and model of the moisture probes? Different probes represent different soil volumes.

Reply to Q13: Yes, we know the make and model of the soil moisture devices and will add the info into the revised text. Surface soil moisture (0–10 cm) was measured using a Theta ML3 handheld probe. For subsurface soil moisture, permanently installed probes were used: SMT-100 sensors (Umwelt-Geräte-Technik GmbH) for forest and grassland sites, and CS650 probes (Campbell Scientific) for agroforestry and crop sites. We will add this information in the Methods section (Lines 226–229) and also update Table S1 (Soil Moisture Measurement Devices and Aggregation Methods) in the Supplementary Material.

The revised text will read:

"Near surface soil moisture (0–10 cm) was measured using a handheld soil moisture device (Theta handheld probe ML3 Sensor) on a monthly basis during two periods of more detailed observations in 2018–2019 and in 2021. Additionally, data were obtained from permanently installed soil moisture probes: SMT-100 at forest and grassland sites, and CS650 at agroforestry and cropland sites. Measurements were recorded at 15-minute intervals with two replicates per depth. To facilitate data

processing and consistency, all soil moisture datasets were aggregated into daily mean values, resulting in one VWC value per site and soil depth. Details of the measurement devices, depth intervals, and aggregation methods is summarized in Table S1."

Q14. Line 239: The abstract reads "simulation of transpiration across forests with different rooting distribution, stand ages, and species compositions." I'm just finished reading through the model description and I'm unclear if the model can simulate different species co-occuring within plots. It sounds like the answer is no. If that's the case, then the line in the abstract may be a little misleading. It can handle species-level rooting distributions, but only if the forest is a monoculture. I understand this is the common condition in many European forests, but in many other parts of the world this is not the case.

Reply to Q14:

We agree that the wording in the abstract may be misleading; apologies for this error. EcoPlot-iso could, in principle, be applied to simulate mixed-species stands, since it operates at the plot scale where a wide range of ecohydrological parameters (e.g., canopy, rooting, and soil properties; see Table S2 can be calibrated at the plot-scale using a Monte Carlo approach, rather than being rigidly tied to a single forest type. However, the scenarios in this study did not simulate multiple species co-occurring within the same stand. Instead, we represented generic monoculture scenarios, where forest type (broadleaf, conifer, agroforestry), stand density, and rooting distributions were varied. We also agree with the reviewer's point that this assumption reflects conditions in many European forests, but may be less representative of regions with high species diversity.

To address these, in the revised manuscript we will:
- clarify in the abstract that transpiration is simulated across generic forest scenarios with contrasting forest types, stand density, and rooting distributions, rather than across mixed-species stands;
- emphasize in the Methods that the current simulations are based on monoculture scenarios; and
- explicitly note this limitation in the Discussion, while also highlighting that EcoPlot-iso could be extended to simulate mixed-species stands in future applications.

Addition to the Discussion (to be included):

"While this study focused on monoculture scenarios (broadleaf, conifer, agroforestry) for clarity and comparability, EcoPlot-iso could in principle be applied to simulate mixed-species stands, as its ecohydrological parameters are calibrated at the plot scale using a Monte Carlo approach. This represents a potential avenue for future development and application, particularly in regions where diverse forest compositions are the norm."

Q15. Line 266: What do you mean "adjusted for three forest types"? If you're testing different forest types influence hydrological fluxes, then this adjustment is going to completely drive the answer. Much more information is needed here.

Reply to Q15:

Thank you for pointing this out; our wording was misleading. What we meant is that the model was driven by observed LAI time series specific to each forest type (broadleaf, coniferous, and agroforestry). As described earlier in the methods, these were derived from the 8-day MODIS LAI product (2000–2024) and then bias-corrected using site measurements (minimum and maximum LAI values) to improve accuracy and reduce noise. This ensured that the LAI inputs reflect realistic seasonal and interannual variability for each land-use type, rather than being artificially tuned. We will revise the text to make this clearer.

The revised text will read:

"For vegetation forcing, we used forest-type–specific observed LAI time series (broadleaf, coniferous, and agroforestry), derived from the MODIS 8-day LAI product (2000–2024) and bias-corrected using site measurements (see Section 3.3; Table 2, Figure S2d)."

Q16. Lines 273 – 278, Scenarios b and c: I don't really understand how you can separate species composition from rooting depths? A number of studies are presenting strong evidence that rooting depths are related to species identity. Another concern is: why do we think rooting distributions are

independent of forest type? It seems you are varying root distributions independently of LAI which doesn't seem correct.

Reply to Q16:
We agree with the reviewer that rooting depths are strongly related to species identity and forest type. In this study, however, our aim was to explore the sensitivity of water partitioning to three key dimensions of forest management characteristics— generic forest type, density, and root distribution— considered separately. To achieve this, forest type was generically represented by type-specific LAI time series, forest density by scaling factors applied to LAI, and forest age by varying rooting depth ($\beta$ parameter), with younger stands represented by shallower rooting and older stands by deeper rooting. We will clarify this modelling assumption in the revised manuscript to emphasise that we are not suggesting rooting depth is independent of species in reality, but rather that this separation was introduced as a scenario design choice for sensitivity testing. Part of the motivation for the study was to illustrate to land managers in our study area how generic choices of land use could affect water use in terms of canopy structure

Q17. Fig. 3: Can you report a metric that is scale dependent? Something like RMSE would help to give a better sense of the model fit. The fit looks good, I just want to pair my subjective interpretation with an objective value.

Reply to Q17:
Thank you for this helpful suggestion. According to your suggestion, we will add a scale-dependent metric in the revised manuscript. Specifically, in addition to the Kling–Gupta Efficiency (KGE), we will report the root mean square error (RMSE) for each simulation in Figure 3 to provide a unit-based measure of model fit.

Q18. Fig. 3: The font size is really small on the y-axis. Can you increase the size?
Reply to Q18: Thank you for pointing this out. We will increase the y-axis font size in Figure 3 to improve readability in the revised manuscript.

Q19. Fig 4: Same issue with font sizes. They are unreadable unless I zoom in to 200% and I'm even very old yet.
Reply to Q19: Thank you for the comment. We will also increase the font sizes in Figure 4 to ensure readability in the revised manuscript.

Q20. Lines 347 – 350: I still don't really understand this LAI scaling. This seems the be the only factor that differentiates the different forest types that are simulated. This is really just an exercise in how sensitive is transpiration to this LAI adjustment. Why were root parameters kept constant across different forest types?

Reply to Q20:
We will clarify that the LAI scaling was implemented to represent differences in forest density rather than forest type per se. As we explained in the reply to Q16, our aim in this study was to explore the impact of water partitioning to three key dimensions of forest management characteristics—generic forest type, density, and root distribution separately. LAI scaling was used to represent forest density varing, species type was represented by type-specific LAI timeseries, and rooting depth ($\beta$ parameter) was varied independently to reflect forest age. We will revise the manuscript to make this modelling assumption clearer.

Q21. Fig 8: This is a good sensitivity test, but its telling me that uncertainty in the rooting depth and LAI scaling factors are driving the results completely.

Reply to Q21:
The aim of this study was to examine how uncertainty in both vegetation structure (represented by the LAI scaling factor) and rooting characteristics (represented by $\beta$ values) influences ecohydrological fluxes across different forest types (captured by the distinct LAI time series for agroforest, broadleaf, and conifer). Figure 8 shows that while rooting depth and LAI scaling factors strongly affect ET, transpiration, and recharge partitioning, these sensitivities vary systematically with forest type. In particular, higher LAI scaling factors generally increase ET and reduce recharge, although the

magnitude of this trade-off differs among forest types. Similarly, rooting depth ($\beta$) modulates the balance between transpiration and recharge, with deeper rooting systems sustaining higher transpiration. Our intention was not only to highlight the role of rooting depth and LAI scaling factors, but also to demonstrate how forest type and canopy density interact with these parameters to shape long-term water partitioning trends. We will clarify this objective (and finding) in the revised manuscript text around Fig. 8 to avoid any impression that the results are driven by a single factor.

Q22. Line 483: There is much more than canopy structure that differentiates different forests from one another. For example, rooting depths likely vary. Beyond that there are different conductance rates, sensitivities to VPD, different tolerances for soil water stress, etc. This study really only considers how these forests might vary in one specific way and presumes forests are monolithic along every other axis. I think this might be too limiting for this study to really tell us about the water fluxes with much confidence.
Reply to Q22:
We agree that forests differ in many additional aspects beyond canopy structure, and would note that we are also evaluating rooting distributions as well. In this study, however, our main intention was to use a parsimonious and generic framework for assessment, rather than to reproduce detailed species-specific physiology. The framework systematically explores variation in three main factors that strongly influence water fluxes at our study sites: canopy density (via LAI scaling factors), rooting distribution (via $\beta$), and forest type (broadleaf, conifer, agroforest, each with distinct LAI time series). These dimensions were selected as they represent key controls on ecohydrological partitioning and allow us to investigate long-term trade-offs between transpiration, evapotranspiration, and recharge. We acknowledge the limitations of this simplified representation and will clarify this in the Discussion, emphasizing that our results should be interpreted as scenario-based insights into generic forest management strategies, rather than detailed predictions for specific species or physiological traits.

Q23. Line 498: I strongly disagree with this logic. These models need more information not because these other models are overly complicated, but rather because the real life system you are trying to model is actually that complicated. This model presented here is probably too simplistic to really capture what is happening. Models should be as simple as possible and no simpler. I speculate that this model has gone too far towards simplicity for the specific question being asked to the point where it doesn't fully inform us.
Reply to Q23:
With respect, there may be a misunderstanding here. We are not arguing for simple models per se, but pointing out that the data needed to drive and evaluate more complex models are rarely available outside research sites. Thus, we agree with the reviewer's perspective that forest ecohydrological systems are inherently complex. Our aim in this study was not to reproduce all species-specific physiological processes, but to provide a parsimonious tracer-aided modelling framework that can quantify relative trade-offs in long-term water partitioning under alternative forest management scenarios in a data-limited, lowland European setting.
To balance simplicity with realism, we (i) dual-calibrated the model against soil moisture and soil-water isotopes using a Monte Carlo, multi-criteria procedure (1,000,000 parameter sets, with the best 100 retained for scenario analyses) to constrain equifinality and ensure transferability across sites, and (ii) explicitly represented rooting distribution with a depth-dependent uptake function ($\beta$) while varying canopy density (LAI scaling) and forest type (broadleaf, conifer, agroforest). These design choices are described in Sections 3.2–3.5 and supported by the skill metrics and uncertainty envelopes presented in Figs. 3–10.
We will revise the sentence around line 498 to clarify that our intention was not to suggest that complex process-based models (e.g., RHESSys, EcH2O) are "overly complicated," but rather that their application can be constrained by high data and computational requirements in regional management contexts. We will also strengthen the Discussion to frame our results as generic, management-relevant insights rather than species-level predictions, and to acknowledge that additional processes (e.g., stomatal conductance, VPD sensitivity, sap-flux or xylem-isotope constraints, lateral flows) remain important directions for future development.

---

## Author Comment (AC2)

**Response to Referee Comment #2:**

Dear editor, reviewer #2,

We sincerely thank the reviewer for the detailed and constructive comments, which will greatly help us to improve the clarity, robustness, and focus of our manuscript. We particularly appreciate the reviewer's recognition of the relevance of isotope-enabled ecohydrological models and the importance of understanding forest management effects on hydrological partitioning.

We recognize that several aspects of the work —particularly the rationale and implementation of our simplified forest-management scenario framework—were not explained sufficiently in the original submission. In revision, we will clarify the new and exploratory nature of the framework and improve it by re-running the modelling to

- (i) distinguish soil from vegetation parameters,
- (ii) keep soil parameters constant while varying vegetation parameters (e.g., LAI, radiation extinction factor, interception capacity parameter), and
- (iii) incorporate forest-specific calibrations to represent broadleaf, conifer, and agroforestry systems more robustly.

These revisions will make the framework much more transparent and physically consistent, while maintaining comparability across scenarios. We believe that the revised version will substantially strengthen the scientific quality and readability of the manuscript, making it suitable for publication in HESS. Below, we provide detailed, point-by-point responses describing how each comment will be addressed in the revised text.

Sincerely,

Dr. Cong Jiang (on behalf of all co-authors)

Dear editors, dear authors,

After careful consideration of the submission I recommend that the manuscript in its present form is not sufficient for publication in HESS.

The described study extended an existing water balance model with a root water uptake (RWU) parametrization from 3 distinct soil layers that is partitioned according to an exponential root distribution function. The authors calibrated four model parametrizations to data of four sites (broadleaf forest, agroforestry, grassland, cropland). At each site seven years of soil moisture and three years of soil water d2H isotopes were available from multiple depths. With the parametrization obtained for broadleaf forest, the authors performed a sensitivity analysis of the model predictions (i.e. of hydrologic partitioning and soil moisture status) to variations in model parameters, namely the seasonal timing and magnitude of LAI (Figure S2d). Furthermore, additional sensitivity analyses further explored the impact of stronger variations in LAI (factors ranging from 0.2 to 1.8) as well as variations in efficiency of root water uptake (parameter beta). As main findings, the authors quantified differences in the water partitioning of yearly available precipitation: highest evapotranspiration (ET) and lowest groundwater recharge (RE) were observed in the model forced with highest LAI and longest growing season (Figure S2d, attributed to coniferous forest). Inversely, lowest ET and highest RE were observed in the model forced with lowest, shortest LAI (Figure S2d, attributed to agroforestry). They quantified the differences between these two model runs on the order of 12% (ET) and 11% (RE) of the yearly precipitation.

Reply: We thank the reviewer for summarizing our approach and main findings so well. The primary goal of this study was to develop a new, parsimonious and generic forest management scenario framework to evaluate how forest type, forest density, and root distribution — associated with forest age—influence long-term water partitioning and ecohydrological resilience under comparable environmental conditions. This new framework was designed to capture the dominant effects of vegetation structure on water partitioning, rather than to reproduce detailed species-specific physiology

**General comments:**

The development of isotope-enabled ecohydrological water balance models with realistic RWU parametrizations is a welcome addition in the field of critical zone hydrology. Such developments are needed to advance our understanding of hydrological partitioning in the critical zone (Guswa 2020). Different model complexities and multiple calibration targets are means to better validate models and reduce model equifinality, thereby leading to mechanistic models that show lower parametric uncertainty (Kuppel 2018, Birkel 2023).

Reply: Yes, we agree, these are exactly our intention (and some of our team have been coauthors of these previous studies).

Having high confidence in the parametrization of model processes is especially crucial when predicting model-derived outputs (such as the hydrologic partitioning) to which the model was not directly calibrated, and which is thus entirely depending on the structural correctness of the model. Understanding hydrological partitioning in the critical zone of forest systems as a function of forest stand properties (land management scenarios) or climate parameters (dry years vs. wet years) is a relevant research problem of importance to forest managers and in scope for HESS.

However, the manuscript in its present form is not sufficient for publication in HESS:

Reply: We thank the reviewer for recognizing the relevance and importance of developing these isotope-enabled ecohydrological models with realistic root water uptake parameterizations to advance understanding of hydrological partitioning in the critical zone.

We are grateful for the reviewer's valuable and constructive feedback and for acknowledging the significance of this research within the scope of HESS.

We have carefully considered all comments and will revise the manuscript accordingly to improve its clarity and scientific rigor. As part of the revision, we will adopt a different approach to the modelling as suggested by the reviewer. We believe that these revisions will substantially enhance the quality of the manuscript, and we hope that the revised version will be considered suitable for publication in HESS.

Q1-1. The approach of extrapolating the model (that was fitted to soil moisture and soil water isotopes at the broadleaf site) to other vegetation types (conifer and agroforestry sites) by simply modifying LAI, while keeping all other model parameters, is not sufficiently substantiated. Even for a "simplified modelling tool" validating the resulting predictions against data from sites containing these vegetation types is required for robust interpretation. Species or plant functional types affect (among others) stomatal control, root distribution, or soil water availability parameters in models (e.g. Cowan 1978, Kuppel 2014, Li 2022, Peters 2025), i.e. processes that are also implicitly present in the RWU parametrization of the EcoPlotiso model (eq. 1-3). The parametrization of these processes should thus likely change when extrapolating the model to other vegetation types.

Reply to Q1-1: We thank the reviewer for summarizing our approach and for the insightful comment on the extrapolation of model parameters between vegetation types (broadleaf, conifer and agroforestry).

Of course, we fully agree that vegetation types differ in functional traits beyond canopy LAI—such as stomatal control, root distribution and soil water availability parameters—that can influence model parameterization. We apologise that we did not make that clearer in the original version. To clarify: vegetation-related processes—spanning canopy interception, evaporation, and root water uptake—are represented in the EcoPlot-iso model by parameters including the leaf area index (LAI), radiation extinction factor (rE), canopy interception storage capacity ( $\alpha$ ), passive interception storage mixing volume (INTp), and the root distribution parameter ( $\beta$ ). Soil-related processes are characterized by parameters such as maximum soil moisture content (Smax, GWmax, Lmax), saturated hydraulic conductivity (k1, k2, k3) and nonlinear scaling parameter (g1, g2, g3) for each soil layer. We will ensure that this is clearly explained in the revised manuscript.

Importantly, the primary goal of this study was to develop a new, parsimonious and generic forest management scenario framework to evaluate how forest type, forest density, and root distribution —associated with forest age—influence long-term water partitioning and ecohydrological resilience under comparable environmental conditions. This framework was designed to capture the dominant effects of vegetation structure—such as interception and transpiration through canopy and root networks—on water partitioning, rather than to reproduce detailed species-specific physiology. To isolate the effects of vegetation characteristics, in the original version, we kept soil parameters constant while vegetation-related parameters, particularly LAI, were varied initially, as LAI strongly controls canopy interception and evapotranspiration partitioning.

However, we acknowledge that any vegetation-type-specific parameterization should also involve other canopy-related parameters (as suggested by the reviewer). Accordingly, in the revised manuscript, we will refine our forest management scenarios framework by incorporating forest-type-specific parameters for broadleaf, conifer, and agroforestry systems. These vegetation parameters (rE,  $\alpha$ , INTp) will be derived from new site-specific calibrations for each forest type, while maintaining soil parameters from the broadleaf site to ensure

comparability (see Figure S8 provided in the response to Q2). Specifically, we will use the median parameter values from 100 behavioural simulations at each site, or apply a cross-combination of the retained parameter ensembles, to represent realistic canopy characteristics across forest types. The calibrated vegetation parameters appear physically consistent, showing median patterns of rE (in absolute magnitude: broadleaf > agroforestry > conifer) and  $\alpha$  (agroforestry

Figure S8. Probability density distributions of the 20 calibrated ecohydrological parameters for five land-use types (broadleaf forest, conifer forest, agroforestry, grassland, and cropland) based on 100 behavioural simulations from the EcoPlot-iso model. Each panel represents one parameter, with kernel density estimates (KDEs) shown in different colours corresponding to each land-use type. Vertical dashed lines indicate the median values of the posterior parameter distributions. Below each subplot, the median values are listed in ascending order (left to right) with text colours matching the respective land-use type. The density plots highlight parameter sensitivities and the distinct parameterization patterns across contrasting vegetation covers.

**Revised Table S2.** EcoPlot-iso parameters, initial and calibrated parameters ranges for calibration. BF: Broadleaf Forest, CF: Conifer Forest, AF: Agroforest, GL: Grassland, CL: Cropland.

| Parameter | D : #                                                            | Sites | T *** 1       | Calibrated range |        |        |  |
|-----------|------------------------------------------------------------------|-------|---------------|------------------|--------|--------|--|
|           | Description                                                      |       | Initial range | Min              | Median | Max    |  |
|           | Radiation extinction factor (dimensionless)                      | BF    | [-0.6, -0.1]  | -0.598           | -0.535 | -0.308 |  |
|           |                                                                  | CF    |               | -0.599           | -0.381 | -0.182 |  |
| rE        |                                                                  | AF    |               | -0.599           | -0.522 | -0.257 |  |
|           |                                                                  | GL    |               | -0.600           | -0.525 | -0.283 |  |
|           |                                                                  | CL    |               | -0.598           | -0.511 | -0.264 |  |
|           | Interception storage capacity parameter (mm per unit of LAI)     | BF    | [0.1, 2.0]    | 0.123            | 1.001  | 1.993  |  |
|           |                                                                  | CF    |               | 0.121            | 0.891  | 1.951  |  |
| $\alpha$  |                                                                  | AF    |               | 0.127            | 0.843  | 1.758  |  |
|           |                                                                  | GL    |               | 0.107            | 0.603  | 1.745  |  |
|           |                                                                  | CL    |               | 0.104            | 0.609  | 1.936  |  |
|           | Maximum soil moisture content in the upper soil compartment (mm) | BF    | [40, 60]      | 40.080           | 48.202 | 59.032 |  |
| Smax      |                                                                  | CF    |               | 40.541           | 48.439 | 57.821 |  |
|           |                                                                  | AF    |               | 41.776           | 53.206 | 59.903 |  |
|           |                                                                  | GL    |               | 40.194           | 47.507 | 59.820 |  |
|           |                                                                  | CL    |               | 40.531           | 53.894 | 59.962 |  |
| Ic        | Soil infiltration capacity (mm/day)                              | BF    | [40, 60]      | 40.163           | 50.559 | 59.845 |  |

| AF                                                                                                                                                                                                                                                                                                                                                                                                                                                                                                                                                                                                                                                                                                                                                                                                                                                                                                                                                                                                                                                                                                                                                                                                                                                                                                                                                                                                                                                                                                                                                                                                                                                                                                                                                                                                                                                                                                                                                                                                                                                                                                                           |                |                                          |    |             |        |        |        |
|------------------------------------------------------------------------------------------------------------------------------------------------------------------------------------------------------------------------------------------------------------------------------------------------------------------------------------------------------------------------------------------------------------------------------------------------------------------------------------------------------------------------------------------------------------------------------------------------------------------------------------------------------------------------------------------------------------------------------------------------------------------------------------------------------------------------------------------------------------------------------------------------------------------------------------------------------------------------------------------------------------------------------------------------------------------------------------------------------------------------------------------------------------------------------------------------------------------------------------------------------------------------------------------------------------------------------------------------------------------------------------------------------------------------------------------------------------------------------------------------------------------------------------------------------------------------------------------------------------------------------------------------------------------------------------------------------------------------------------------------------------------------------------------------------------------------------------------------------------------------------------------------------------------------------------------------------------------------------------------------------------------------------------------------------------------------------------------------------------------------------|----------------|------------------------------------------|----|-------------|--------|--------|--------|
| Saturated hydraulic conductivity of the upper soil compartment (mm/day)                                                                                                                                                                                                                                                                                                                                                                                                                                                                                                                                                                                                                                                                                                                                                                                                                                                                                                                                                                                                                                                                                                                                                                                                                                                                                                                                                                                                                                                                                                                                                                                                                                                                                                                                                                                                                                                                                                                                                                                                                                                      |                |                                          |    |             |        |        |        |
| Saturated hydraulic conductivity of the upper soil compartment (mm/day)                                                                                                                                                                                                                                                                                                                                                                                                                                                                                                                                                                                                                                                                                                                                                                                                                                                                                                                                                                                                                                                                                                                                                                                                                                                                                                                                                                                                                                                                                                                                                                                                                                                                                                                                                                                                                                                                                                                                                                                                                                                      |                |                                          |    |             |        |        |        |
| Saturated hydraulic conductivity of the upper soil compartment (mm/day)                                                                                                                                                                                                                                                                                                                                                                                                                                                                                                                                                                                                                                                                                                                                                                                                                                                                                                                                                                                                                                                                                                                                                                                                                                                                                                                                                                                                                                                                                                                                                                                                                                                                                                                                                                                                                                                                                                                                                                                                                                                      |                |                                          |    | -           |        |        |        |
| Saturated hydraulic conductivity of the upper soil compartment (mm/day)                                                                                                                                                                                                                                                                                                                                                                                                                                                                                                                                                                                                                                                                                                                                                                                                                                                                                                                                                                                                                                                                                                                                                                                                                                                                                                                                                                                                                                                                                                                                                                                                                                                                                                                                                                                                                                                                                                                                                                                                                                                      |                |                                          |    |             |        |        |        |
| Saturated hydraulic conductivity of the lower soil compartment (mm/day)                                                                                                                                                                                                                                                                                                                                                                                                                                                                                                                                                                                                                                                                                                                                                                                                                                                                                                                                                                                                                                                                                                                                                                                                                                                                                                                                                                                                                                                                                                                                                                                                                                                                                                                                                                                                                                                                                                                                                                                                                                                      |                |                                          |    | -           |        |        |        |
| Lange   Soliton   Solito | Ira I          |                                          |    | [1 20]      |        |        |        |
| Saturated hydraulic conductivity of the lower soil compartment (mm/day)                                                                                                                                                                                                                                                                                                                                                                                                                                                                                                                                                                                                                                                                                                                                                                                                                                                                                                                                                                                                                                                                                                                                                                                                                                                                                                                                                                                                                                                                                                                                                                                                                                                                                                                                                                                                                                                                                                                                                                                                                                                      | KS I           | upper soil compartment (mm/day)          |    | [1, 20]     |        |        |        |
| Saturated hydraulic conductivity of the lower soil compartment (mm/day)                                                                                                                                                                                                                                                                                                                                                                                                                                                                                                                                                                                                                                                                                                                                                                                                                                                                                                                                                                                                                                                                                                                                                                                                                                                                                                                                                                                                                                                                                                                                                                                                                                                                                                                                                                                                                                                                                                                                                                                                                                                      |                |                                          |    |             |        |        |        |
| Saturated hydraulic conductivity of the lower soil compartment (mm/day)                                                                                                                                                                                                                                                                                                                                                                                                                                                                                                                                                                                                                                                                                                                                                                                                                                                                                                                                                                                                                                                                                                                                                                                                                                                                                                                                                                                                                                                                                                                                                                                                                                                                                                                                                                                                                                                                                                                                                                                                                                                      |                |                                          |    |             |        |        |        |
| Saturated hydraulic conductivity of the lower soil compartment (minday)                                                                                                                                                                                                                                                                                                                                                                                                                                                                                                                                                                                                                                                                                                                                                                                                                                                                                                                                                                                                                                                                                                                                                                                                                                                                                                                                                                                                                                                                                                                                                                                                                                                                                                                                                                                                                                                                                                                                                                                                                                                      |                |                                          | -  | [1, 20]     |        |        |        |
| Solution   Color   1,005   1,006   19,193   19,858   19,858   18,858   18,858   18,858   18,858   18,858   18,858   18,858   18,859   13,291   1,004   1,004   1,004   1,004   1,004   1,004   1,004   1,004   1,004   1,004   1,004   1,004   1,004   1,004   1,004   1,004   1,004   1,004   1,004   1,004   1,004   1,004   1,004   1,004   1,004   1,004   1,004   1,004   1,004   1,004   1,004   1,004   1,004   1,004   1,004   1,004   1,004   1,004   1,004   1,004   1,004   1,004   1,004   1,004   1,004   1,004   1,004   1,004   1,004   1,004   1,004   1,004   1,004   1,004   1,004   1,004   1,004   1,004   1,004   1,004   1,004   1,004   1,004   1,004   1,004   1,004   1,004   1,004   1,004   1,004   1,004   1,004   1,004   1,004   1,004   1,004   1,004   1,004   1,004   1,004   1,004   1,004   1,004   1,004   1,004   1,004   1,004   1,004   1,004   1,004   1,004   1,004   1,004   1,004   1,004   1,004   1,004   1,004   1,004   1,004   1,004   1,004   1,004   1,004   1,004   1,004   1,004   1,004   1,004   1,004   1,004   1,004   1,004   1,004   1,004   1,004   1,004   1,004   1,004   1,004   1,004   1,004   1,004   1,004   1,004   1,004   1,004   1,004   1,004   1,004   1,004   1,004   1,004   1,004   1,004   1,004   1,004   1,004   1,004   1,004   1,004   1,004   1,004   1,004   1,004   1,004   1,004   1,004   1,004   1,004   1,004   1,004   1,004   1,004   1,004   1,004   1,004   1,004   1,004   1,004   1,004   1,004   1,004   1,004   1,004   1,004   1,004   1,004   1,004   1,004   1,004   1,004   1,004   1,004   1,004   1,004   1,004   1,004   1,004   1,004   1,004   1,004   1,004   1,004   1,004   1,004   1,004   1,004   1,004   1,004   1,004   1,004   1,004   1,004   1,004   1,004   1,004   1,004   1,004   1,004   1,004   1,004   1,004   1,004   1,004   1,004   1,004   1,004   1,004   1,004   1,004   1,004   1,004   1,004   1,004   1,004   1,004   1,004   1,004   1,004   1,004   1,004   1,004   1,004   1,004   1,004   1,004   1,004   1,004   1,004   1,004   1,004   1,004   1,004   1,004   1,004   1,004   1,004 | ks2            |                                          |    |             |        |        |        |
| Saturated hydraulic conductivity of the deeper soil compartment (mm/day)                                                                                                                                                                                                                                                                                                                                                                                                                                                                                                                                                                                                                                                                                                                                                                                                                                                                                                                                                                                                                                                                                                                                                                                                                                                                                                                                                                                                                                                                                                                                                                                                                                                                                                                                                                                                                                                                                                                                                                                                                                                     |                | lower soil compartment (mm/day)          |    | [-,]        |        |        |        |
| Saturated hydraulic conductivity of the deeper soil compartment (mm/day)                                                                                                                                                                                                                                                                                                                                                                                                                                                                                                                                                                                                                                                                                                                                                                                                                                                                                                                                                                                                                                                                                                                                                                                                                                                                                                                                                                                                                                                                                                                                                                                                                                                                                                                                                                                                                                                                                                                                                                                                                                                     |                |                                          |    | 1           |        |        |        |
| Salurate hydraulic conductivity of the deeper soil compartment (mm/day)                                                                                                                                                                                                                                                                                                                                                                                                                                                                                                                                                                                                                                                                                                                                                                                                                                                                                                                                                                                                                                                                                                                                                                                                                                                                                                                                                                                                                                                                                                                                                                                                                                                                                                                                                                                                                                                                                                                                                                                                                                                      |                |                                          | BF | [1, 20]     |        | 3.809  |        |
| AF                                                                                                                                                                                                                                                                                                                                                                                                                                                                                                                                                                                                                                                                                                                                                                                                                                                                                                                                                                                                                                                                                                                                                                                                                                                                                                                                                                                                                                                                                                                                                                                                                                                                                                                                                                                                                                                                                                                                                                                                                                                                                                                           |                |                                          | CF | [50, 100]   | 51.940 | 75.149 | 99.879 |
| Maximum soil moisture content in the lower soil compartment (mm)   BF   CF   1,20   1,347   9,655   19,841                                                                                                                                                                                                                                                                                                                                                                                                                                                                                                                                                                                                                                                                                                                                                                                                                                                                                                                                                                                                                                                                                                                                                                                                                                                                                                                                                                                                                                                                                                                                                                                                                                                                                                                                                                                                                                                                                                                                                                                                                   | ks3            |                                          | AF | [1, 20]     | 1.017  | 3.831  | 11.877 |
| ## Assimation of the lower soil compartment (mm)   Fig.   CF   So. 1058   65.115   95.84                                                                                                                                                                                                                                                                                                                                                                                                                                                                                                                                                                                                                                                                                                                                                                                                                                                                                                                                                                                                                                                                                                                                                                                                                                                                                                                                                                                                                                                                                                                                                                                                                                                                                                                                                                                                                                                                                                                                                                                                                                     |                | deeper son compartment (mm/day)          | GL | [1, 20]     | 1.045  | 4.245  | 10.895 |
| Maximum soil moisture content in the lower soil compartment (mm)                                                                                                                                                                                                                                                                                                                                                                                                                                                                                                                                                                                                                                                                                                                                                                                                                                                                                                                                                                                                                                                                                                                                                                                                                                                                                                                                                                                                                                                                                                                                                                                                                                                                                                                                                                                                                                                                                                                                                                                                                                                             |                |                                          | CL | [1, 20]     | 1.337  | 9.655  | 19.841 |
| Maximum soil moisture content in the lower soil compartment (mm)                                                                                                                                                                                                                                                                                                                                                                                                                                                                                                                                                                                                                                                                                                                                                                                                                                                                                                                                                                                                                                                                                                                                                                                                                                                                                                                                                                                                                                                                                                                                                                                                                                                                                                                                                                                                                                                                                                                                                                                                                                                             |                |                                          | BF |             | 50.058 | 65.115 | 95.584 |
| Lmax                                                                                                                                                                                                                                                                                                                                                                                                                                                                                                                                                                                                                                                                                                                                                                                                                                                                                                                                                                                                                                                                                                                                                                                                                                                                                                                                                                                                                                                                                                                                                                                                                                                                                                                                                                                                                                                                                                                                                                                                                                                                                                                         |                | Maximum sail maisture content in the     | CF |             | 50.180 | 63.310 | 98.699 |
| ## Action                                                                                                                                                                                                                                                                                                                                                                                                                                                                                                                                                                                                                                                                                                                                                                                                                                                                                                                                                                                                                                                                                                                                                                                                                                                                                                                                                                                                                                                                                                                                                                                                                                                                                                                                                                                                                                                                                                                                                                                                                                                                                                                    | GWmax          |                                          |    | [50, 100]   |        |        |        |
| Maximum soil moisture content in the deeper soil compartment (mm)                                                                                                                                                                                                                                                                                                                                                                                                                                                                                                                                                                                                                                                                                                                                                                                                                                                                                                                                                                                                                                                                                                                                                                                                                                                                                                                                                                                                                                                                                                                                                                                                                                                                                                                                                                                                                                                                                                                                                                                                                                                            |                | lower som compartment (mm)               | GL |             |        |        |        |
| ### Maximum soil moisture content in the deeper soil compartment (mm)  ### AF [250,450] 250,178 314,194 447,936 GL 1100,300] 100,485 163,030 295,118                                                                                                                                                                                                                                                                                                                                                                                                                                                                                                                                                                                                                                                                                                                                                                                                                                                                                                                                                                                                                                                                                                                                                                                                                                                                                                                                                                                                                                                                                                                                                                                                                                                                                                                                                                                                                                                                                                                                                                         |                |                                          |    |             |        |        |        |
| ### AF   250, 450   250, 178   314, 194   447, 936                                                                                                                                                                                                                                                                                                                                                                                                                                                                                                                                                                                                                                                                                                                                                                                                                                                                                                                                                                                                                                                                                                                                                                                                                                                                                                                                                                                                                                                                                                                                                                                                                                                                                                                                                                                                                                                                                                                                                                                                                                                                           |                |                                          |    |             |        |        |        |
| ## deeper soil compartment (mm)                                                                                                                                                                                                                                                                                                                                                                                                                                                                                                                                                                                                                                                                                                                                                                                                                                                                                                                                                                                                                                                                                                                                                                                                                                                                                                                                                                                                                                                                                                                                                                                                                                                                                                                                                                                                                                                                                                                                                                                                                                                                                              |                | Maximum soil moisture content in the     |    |             |        |        |        |
| StoSo   Passive upper soil storage mixing volume (mm)   Passive lower soil storage mixing volume (mm)   Passive deep  | Lmax           |                                          |    |             |        |        |        |
| StoSo   Passive upper soil storage mixing volume (mm)   Passive lower soil storage mixing volume (mm)   Passive deep soil storage mixing volume (mm)   Passive  |                | 1                                        | -  |             |        |        |        |
| StoSo   Passive upper soil storage mixing volume (mm)   Passive lower soil storage mixing volume (mm)   Passive deep  |                |                                          |    | [250, 450]  |        |        |        |
| StoSo   Passive upper soil storage mixing volume (mm)   Passive lower soil storage mixing volume (mm)   Passive deep  |                |                                          |    |             |        |        |        |
| Soli compartment                                                                                                                                                                                                                                                                                                                                                                                                                                                                                                                                                                                                                                                                                                                                                                                                                                                                                                                                                                                                                                                                                                                                                                                                                                                                                                                                                                                                                                                                                                                                                                                                                                                                                                                                                                                                                                                                                                                                                                                                                                                                                                             | - 1            |                                          |    | F1 - F1     |        |        |        |
| StoSo   Passive upper soil storage mixing volume (mm)   Passive upper soil storage mixing volume (mm)   Passive upper soil storage mixing volume (mm)   Passive deep | g1             |                                          |    | [1, 5]      |        |        |        |
| StoSo   Passive upper soil storage mixing volume (mm)   Passive upper soil storage mixing volume (mm)   Passive upper soil storage mixing volume (mm)   Passive deep |                |                                          |    | +           |        |        |        |
| Solution   Solution  |                |                                          |    |             |        |        |        |
| Solution   Solution  |                |                                          |    |             |        |        |        |
| Soil compartment                                                                                                                                                                                                                                                                                                                                                                                                                                                                                                                                                                                                                                                                                                                                                                                                                                                                                                                                                                                                                                                                                                                                                                                                                                                                                                                                                                                                                                                                                                                                                                                                                                                                                                                                                                                                                                                                                                                                                                                                                                                                                                             | α2             |                                          |    | [1.5]       |        |        |        |
| StoSo   Passive interception storage mixing volume (mm)   Passive upper soil storage mixing volume (mm)   Passive lower soil storage mixing volume (mm)   Passive deep soil storage mixing volume (mm)   Passive | 82             | soil compartment                         |    | [1, 3]      |        |        |        |
| Nonlinear scaling parameter for the deeper soil compartment                                                                                                                                                                                                                                                                                                                                                                                                                                                                                                                                                                                                                                                                                                                                                                                                                                                                                                                                                                                                                                                                                                                                                                                                                                                                                                                                                                                                                                                                                                                                                                                                                                                                                                                                                                                                                                                                                                                                                                                                                                                                  |                |                                          |    | 1           |        |        |        |
| Nonlinear scaling parameter for the deeper soil compartment                                                                                                                                                                                                                                                                                                                                                                                                                                                                                                                                                                                                                                                                                                                                                                                                                                                                                                                                                                                                                                                                                                                                                                                                                                                                                                                                                                                                                                                                                                                                                                                                                                                                                                                                                                                                                                                                                                                                                                                                                                                                  |                |                                          | 1  |             |        |        |        |
| Nonlinear scaling parameter for the deeper soil compartment                                                                                                                                                                                                                                                                                                                                                                                                                                                                                                                                                                                                                                                                                                                                                                                                                                                                                                                                                                                                                                                                                                                                                                                                                                                                                                                                                                                                                                                                                                                                                                                                                                                                                                                                                                                                                                                                                                                                                                                                                                                                  |                |                                          |    | -           |        |        |        |
| Preferential flow path parameter (dimensionless)                                                                                                                                                                                                                                                                                                                                                                                                                                                                                                                                                                                                                                                                                                                                                                                                                                                                                                                                                                                                                                                                                                                                                                                                                                                                                                                                                                                                                                                                                                                                                                                                                                                                                                                                                                                                                                                                                                                                                                                                                                                                             | 23      |                                          | AF | [1, 5]      |        | 2.469  |        |
| Preferential flow path parameter (dimensionless)                                                                                                                                                                                                                                                                                                                                                                                                                                                                                                                                                                                                                                                                                                                                                                                                                                                                                                                                                                                                                                                                                                                                                                                                                                                                                                                                                                                                                                                                                                                                                                                                                                                                                                                                                                                                                                                                                                                                                                                                                                                                             | 8-             |                                          |    |             |        |        |        |
| Preferential flow path parameter (dimensionless)                                                                                                                                                                                                                                                                                                                                                                                                                                                                                                                                                                                                                                                                                                                                                                                                                                                                                                                                                                                                                                                                                                                                                                                                                                                                                                                                                                                                                                                                                                                                                                                                                                                                                                                                                                                                                                                                                                                                                                                                                                                                             |                |                                          | CL | 1           | 2.235  | 3.993  | 4.979  |
| Preferential flow path parameter (dimensionless)                                                                                                                                                                                                                                                                                                                                                                                                                                                                                                                                                                                                                                                                                                                                                                                                                                                                                                                                                                                                                                                                                                                                                                                                                                                                                                                                                                                                                                                                                                                                                                                                                                                                                                                                                                                                                                                                                                                                                                                                                                                                             |                |                                          | BF |             | 0.217  | 0.618  | 0.880  |
| IntSp                                                                                                                                                                                                                                                                                                                                                                                                                                                                                                                                                                                                                                                                                                                                                                                                                                                                                                                                                                                                                                                                                                                                                                                                                                                                                                                                                                                                                                                                                                                                                                                                                                                                                                                                                                                                                                                                                                                                                                                                                                                                                                                        |                |                                          | CF |             |        | 0.271  |        |
| Passive interception storage mixing volume (mm)                                                                                                                                                                                                                                                                                                                                                                                                                                                                                                                                                                                                                                                                                                                                                                                                                                                                                                                                                                                                                                                                                                                                                                                                                                                                                                                                                                                                                                                                                                                                                                                                                                                                                                                                                                                                                                                                                                                                                                                                                                                                              | PFScale |                                          | AF | [0.1, 0.9]  | 0.101  | 0.302  | 0.755  |
| Passive interception storage mixing volume (mm)                                                                                                                                                                                                                                                                                                                                                                                                                                                                                                                                                                                                                                                                                                                                                                                                                                                                                                                                                                                                                                                                                                                                                                                                                                                                                                                                                                                                                                                                                                                                                                                                                                                                                                                                                                                                                                                                                                                                                                                                                                                                              |                |                                          | GL |             | 0.105  | 0.460  | 0.806  |
| Passive interception storage mixing volume (mm)                                                                                                                                                                                                                                                                                                                                                                                                                                                                                                                                                                                                                                                                                                                                                                                                                                                                                                                                                                                                                                                                                                                                                                                                                                                                                                                                                                                                                                                                                                                                                                                                                                                                                                                                                                                                                                                                                                                                                                                                                                                                              |                |                                          |    |             |        |        | 0.715  |
| StoSo                                                                                                                                                                                                                                                                                                                                                                                                                                                                                                                                                                                                                                                                                                                                                                                                                                                                                                                                                                                                                                                                                                                                                                                                                                                                                                                                                                                                                                                                                                                                                                                                                                                                                                                                                                                                                                                                                                                                                                                                                                                                                                                        |                |                                          |    |             |        |        |        |
| AF         [0.3, 1]         0.306         0.760         0.998           StoSo         CL         0.506         0.751         0.993           BF         3.173         14.941         19.809           1.364         9.220         19.236           1.048         3.964         12.078           1.003         4.372         15.261           BF         3.159         11.427         27.790           1.093         4.372         15.261           AF         [3, 40]         3.383         13.983         36.908           3.006         10.503         25.603           3.751         15.748         35.511           BF         32.718         78.886         98.972           CF         10.304         30.913         84.782           AF         [10, 100]         11.818         42.310         97.798           BF         [10, 100]         11.818         42.310         97.798           BF         [10, 600         46.176         99.389                                                                                                                                                                                                                                                                                                                                                                                                                                                                                                                                                                                                                                                                                                                                                                                                                                                                                                                                                                                                                                                                                                                                                                        |                |                                          |    |             |        |        |        |
| StoSo   Passive upper soil storage mixing volume (mm)   Passive lower soil storage mixing volume (mm)   Passive lower soil storage mixing volume (mm)   Passive lower soil storage mixing volume (mm)   Passive deep soil storage mixing volume (mm)   Passive dee | IntSp          |                                          |    | [0.5, 1]    |        |        |        |
| Passive upper soil storage mixing volume (mm)   BF   CF   I   1.364   9.220   19.236   1.089   4.296   16.457   1.048   3.964   12.078   1.003   4.372   15.261   1.003   4.372   15.261   1.003   4.372   15.261   1.052   1.052   1.052   1.052   1.052   1.052   1.052   1.052   1.052   1.052   1.052   1.052   1.052   1.052   1.052   1.052   1.052   1.052   1.052   1.052   1.052   1.052   1.052   1.052   1.052   1.052   1.052   1.052   1.052   1.052   1.052   1.052   1.052   1.052   1.052   1.052   1.052   1.052   1.052   1.052   1.052   1.052   1.052   1.052   1.052   1.052   1.052   1.052   1.052   1.052   1.052   1.052   1.052   1.052   1.052   1.052   1.052   1.052   1.052   1.052   1.052   1.052   1.052   1.052   1.052   1.052   1.052   1.052   1.052   1.052   1.052   1.052   1.052   1.052   1.052   1.052   1.052   1.052   1.052   1.052   1.052   1.052   1.052   1.052   1.052   1.052   1.052   1.052   1.052   1.052   1.052   1.052   1.052   1.052   1.052   1.052   1.052   1.052   1.052   1.052   1.052   1.052   1.052   1.052   1.052   1.052   1.052   1.052   1.052   1.052   1.052   1.052   1.052   1.052   1.052   1.052   1.052   1.052   1.052   1.052   1.052   1.052   1.052   1.052   1.052   1.052   1.052   1.052   1.052   1.052   1.052   1.052   1.052   1.052   1.052   1.052   1.052   1.052   1.052   1.052   1.052   1.052   1.052   1.052   1.052   1.052   1.052   1.052   1.052   1.052   1.052   1.052   1.052   1.052   1.052   1.052   1.052   1.052   1.052   1.052   1.052   1.052   1.052   1.052   1.052   1.052   1.052   1.052   1.052   1.052   1.052   1.052   1.052   1.052   1.052   1.052   1.052   1.052   1.052   1.052   1.052   1.052   1.052   1.052   1.052   1.052   1.052   1.052   1.052   1.052   1.052   1.052   1.052   1.052   1.052   1.052   1.052   1.052   1.052   1.052   1.052   1.052   1.052   1.052   1.052   1.052   1.052   1.052   1.052   1.052   1.052   1.052   1.052   1.052   1.052   1.052   1.052   1.052   1.052   1.052   1.052   1.052   1.052   1.052   1.052   1.052   1.052   1.052   1.052    |                |                                          |    |             |        |        |        |
| Passive upper soil storage mixing volume (mm)                                                                                                                                                                                                                                                                                                                                                                                                                                                                                                                                                                                                                                                                                                                                                                                                                                                                                                                                                                                                                                                                                                                                                                                                                                                                                                                                                                                                                                                                                                                                                                                                                                                                                                                                                                                                                                                                                                                                                                                                                                                                                |                |                                          |    |             |        |        |        |
| Passive upper soil storage mixing volume (mm)                                                                                                                                                                                                                                                                                                                                                                                                                                                                                                                                                                                                                                                                                                                                                                                                                                                                                                                                                                                                                                                                                                                                                                                                                                                                                                                                                                                                                                                                                                                                                                                                                                                                                                                                                                                                                                                                                                                                                                                                                                                                                |                |                                          |    | -           |        |        |        |
| Solution   Column   | A -            | Passive upper soil storage mixing volume |    | F1 003      |        |        |        |
| Passive lower soil storage mixing volume (mm)                                                                                                                                                                                                                                                                                                                                                                                                                                                                                                                                                                                                                                                                                                                                                                                                                                                                                                                                                                                                                                                                                                                                                                                                                                                                                                                                                                                                                                                                                                                                                                                                                                                                                                                                                                                                                                                                                                                                                                                                                                                                                | StoSo          | 11                                       |    | [1, 20]     |        |        |        |
| Passive lower soil storage mixing volume (mm)   BF                                                                                                                                                                                                                                                                                                                                                                                                                                                                                                                                                                                                                                                                                                                                                                                                                                                                                                                                                                                                                                                                                                                                                                                                                                                                                                                                                                                                                                                                                                                                                                                                                                                                                                                                                                                                                                                                                                                                                                                                                                                                           |                |                                          |    |             |        |        |        |
| Passive lower soil storage mixing volume (mm)                                                                                                                                                                                                                                                                                                                                                                                                                                                                                                                                                                                                                                                                                                                                                                                                                                                                                                                                                                                                                                                                                                                                                                                                                                                                                                                                                                                                                                                                                                                                                                                                                                                                                                                                                                                                                                                                                                                                                                                                                                                                                |                |                                          |    |             |        |        |        |
| Passive lower soil storage mixing volume (mm)                                                                                                                                                                                                                                                                                                                                                                                                                                                                                                                                                                                                                                                                                                                                                                                                                                                                                                                                                                                                                                                                                                                                                                                                                                                                                                                                                                                                                                                                                                                                                                                                                                                                                                                                                                                                                                                                                                                                                                                                                                                                                |                |                                          |    | 1           |        |        |        |
| CL   Seasonality factor in the Craig-Gordon   CR   CS   CS   CS   CS   CS   CS   CS                                                                                                                                                                                                                                                                                                                                                                                                                                                                                                                                                                                                                                                                                                                                                                                                                                                                                                                                                                                                                                                                                                                                                                                                                                                                                                                                                                                                                                                                                                                                                                                                                                                                                                                                                                                                                                                                                                                                                                                                                                          | gwSp           |                                          |    | [3 40]      |        |        |        |
| DowSP   Passive deep soil storage mixing volume (mm)   CL   BF   32.718   78.886   98.972                                                                                                                                                                                                                                                                                                                                                                                                                                                                                                                                                                                                                                                                                                                                                                                                                                                                                                                                                                                                                                                                                                                                                                                                                                                                                                                                                                                                                                                                                                                                                                                                                                                                                                                                                                                                                                                                                                                                                                                                                                    |                |                                          |    | [3,40]      |        |        |        |
| Passive deep soil storage mixing volume (mm)   BF   CF   10.304   30.913   84.782                                                                                                                                                                                                                                                                                                                                                                                                                                                                                                                                                                                                                                                                                                                                                                                                                                                                                                                                                                                                                                                                                                                                                                                                                                                                                                                                                                                                                                                                                                                                                                                                                                                                                                                                                                                                                                                                                                                                                                                                                                            |                |                                          |    |             |        |        |        |
| Passive deep soil storage mixing volume (mm)                                                                                                                                                                                                                                                                                                                                                                                                                                                                                                                                                                                                                                                                                                                                                                                                                                                                                                                                                                                                                                                                                                                                                                                                                                                                                                                                                                                                                                                                                                                                                                                                                                                                                                                                                                                                                                                                                                                                                                                                                                                                                 |                |                                          |    |             |        |        |        |
| Passive deep soil storage mixing volume (mm)                                                                                                                                                                                                                                                                                                                                                                                                                                                                                                                                                                                                                                                                                                                                                                                                                                                                                                                                                                                                                                                                                                                                                                                                                                                                                                                                                                                                                                                                                                                                                                                                                                                                                                                                                                                                                                                                                                                                                                                                                                                                                 | lowSP          |                                          |    | [10, 100]   |        |        |        |
| GL 10.906 58.591 98.924 CL 10.600 46.176 99.389  Begin to the Craig-Gordon BF 10.25 0.91 0.262 0.587 0.899                                                                                                                                                                                                                                                                                                                                                                                                                                                                                                                                                                                                                                                                                                                                                                                                                                                                                                                                                                                                                                                                                                                                                                                                                                                                                                                                                                                                                                                                                                                                                                                                                                                                                                                                                                                                                                                                                                                                                                                                                   |                |                                          |    |             |        |        |        |
| CL 10.600 46.176 99.389  Seasonality factor in the Craig-Gordon BF 10.25 0.01 0.262 0.587 0.899                                                                                                                                                                                                                                                                                                                                                                                                                                                                                                                                                                                                                                                                                                                                                                                                                                                                                                                                                                                                                                                                                                                                                                                                                                                                                                                                                                                                                                                                                                                                                                                                                                                                                                                                                                                                                                                                                                                                                                                                                              |                |                                          |    |             |        |        |        |
| Seasonality factor in the Craig-Gordon BF [0.25, 0.01] 0.262 0.587 0.899                                                                                                                                                                                                                                                                                                                                                                                                                                                                                                                                                                                                                                                                                                                                                                                                                                                                                                                                                                                                                                                                                                                                                                                                                                                                                                                                                                                                                                                                                                                                                                                                                                                                                                                                                                                                                                                                                                                                                                                                                                                     |                |                                          |    |             |        |        |        |
|                                                                                                                                                                                                                                                                                                                                                                                                                                                                                                                                                                                                                                                                                                                                                                                                                                                                                                                                                                                                                                                                                                                                                                                                                                                                                                                                                                                                                                                                                                                                                                                                                                                                                                                                                                                                                                                                                                                                                                                                                                                                                                                              | k              | Seasonality factor in the Craig-Gordon   |    | [0.25, 0.9] |        |        |        |
|                                                                                                                                                                                                                                                                                                                                                                                                                                                                                                                                                                                                                                                                                                                                                                                                                                                                                                                                                                                                                                                                                                                                                                                                                                                                                                                                                                                                                                                                                                                                                                                                                                                                                                                                                                                                                                                                                                                                                                                                                                                                                                                              |                | model (dimensionless)                    | CF |             | 0.262  | 0.578  | 0.899  |

|   |                                                                        | AF |              | 0.251 | 0.537 | 0.898 |
|---|------------------------------------------------------------------------|----|--------------|-------|-------|-------|
|   |                                                                        | GL |              | 0.250 | 0.639 | 0.885 |
|   |                                                                        | CL |              | 0.265 | 0.627 | 0.896 |
|   |                                                                        | BF |              | 0.251 | 0.510 | 0.749 |
|   | Water vapor mixing ratio in the Craig-
Gordon model (dimensionless) | CF | [0.25, 0.75] | 0.266 | 0.508 | 0.745 |
| x |                                                                        | AF |              | 0.251 | 0.506 | 0.744 |
|   |                                                                        | GL |              | 0.251 | 0.495 | 0.748 |
|   |                                                                        | CL |              | 0.252 | 0.454 | 0.748 |
|   | Root distribution factor (dimensionless)                               | BF |              | 0.009 | 0.443 | 1.423 |
| β |                                                                        | CF |              | 0.001 | 0.610 | 1.967 |
|   |                                                                        | AF | [0, 2]       | 0.004 | 0.279 | 1.514 |
|   |                                                                        | GL | ]            | 0.029 | 0.468 | 1.698 |
|   |                                                                        | CL |              | 0.007 | 0.781 | 1.973 |

**Revised Table 3.** Kling–Gupta Efficiency (KGE) values for soil moisture and soil water isotopes ( $\delta^2$ H), comparing observed and mean simulated values at each land-use site.

| Sites               | Soil moisture          |                        |                       | Soil water isotope δ 2 H |                        |                       |  |  |
|---------------------|------------------------|------------------------|-----------------------|-------------------------------------|------------------------|-----------------------|--|--|
|                     | Upper soil compartment | Lower soil compartment | Deep soil compartment | Upper soil compartment              | Lower soil compartment | Deep soil compartment |  |  |
| Broadleaf
Forest | 0.60                   | 0.69                   | 0.78                  | 0.58                                | 0.74                   | 0.62                  |  |  |
| Conifer forest      | 0.59                   | 0.60                   | 0.67                  | 0.68                                | 0.81                   | 0.52                  |  |  |
| Agroforestry        | 0.72                   | 0.76                   | 0.78                  | 0.81                                | 0.84                   | 0.78                  |  |  |
| Grassland           | 0.86                   | 0.69                   | 0.71                  | 0.72                                | 0.77                   | 0.59                  |  |  |
| Cropland            | 0.46                   | 0.61                   | 0.73                  | 0.82                                | 0.84                   | 0.31                  |  |  |

Q3. Eventually the manuscript shows many further (sometimes even redundant, e.g. Fig 7/8) model outputs. I struggled as a reader to understand the decision to show that many. Here less might be more. These figures illustrate the results of varying the two other "management dimensions", for which I believe a "model sensitivity analysis" would be a clearer terminology. Reply to Q3: We thank the reviewer for this valuable observation. We agree that some figures (e.g., Figs. 7, 8) might be redundant and that a clearer presentation would strengthen the paper. In the revised manuscript, as suggested, we will streamline the results by removing Figs. 4b–4c, as Fig. 4d already represents the sum of 4b and 4c (as also noted in Q7) and by moving some of the less central visualizations (e.g., current Figs. 7 and 10) to the Supporting Information.

We acknowledge that our multi-dimensional forest management framework—varying forest type, canopy density, and root distribution—may resemble a model sensitivity analysis in structure. However, importantly, our intention was to use these controlled variations as a generic scenario experiment to isolate the dominant vegetation controls on water partitioning, rather than to quantify formal model parameter sensitivity (further explained in Q4). This approach aligns with the study's main goal of developing a parsimonious and generic forest management framework to assess how forest type, canopy structure, and rooting depth influence long-term ecohydrological dynamics under comparable environmental conditions. However, it seems we did not present this sufficiently well. We will clarify this conceptual distinction and terminology in the revised manuscript.

Q4-1. The parameters chosen for this sensitivity analysis of the model (i.e. LAI and beta) have rather straightforward effects on partitioning: increasing LAI and decreasing beta both increase ET relative to RE. The directions (although not the magnitudes) of these effects can be straightforwardly derived from the model formulation: Essentially, water partitioning in the model is driven by the efficiency of different fluxes to access the freshly fallen (and intercepted) or soil-stored precipitation water. Figure 4 illustrates the eventual fate of that water: a) evaporation from canopy, soil evaporation and transpiration (ET), b) groundwater recharge

(RE), c) surface runoff (Qs), or d) change in soil storage. We can reduce the options, given that Qs is negligibly small in this broadleaf forest site (Figure 4). Further, ignoring storage change by assuming zero change on yearly time scales leaves us with two remaining options: a) ET or b) RE. Thus any model change that improves efficiency of interception (e.g. larger LAI, Eq. 4 in Stevenson 2023), evaporation, or transpiration (e.g. smaller beta, Eq. 1-3 present manuscript, or larger LAI Eq. 4 in Stevenson 2023) favours ET instead of RE. Only water that is neither intercepted, evaporated nor transpired can eventually become groundwater recharge. Similar argumentation can be made for the fraction Transpiration/ET. This argumentation summarises most of the directions of the trends shown in Figures 7,8,9,10. It is true that the performed sensitivity analysis, however, was able to quantify \*\*magnitudes\*\* of these effects with the chosen model parameterization. However, also note that these magnitudes strongly depend on the chosen range of parameter variation. They appeared to be chosen without clear justification as 0.2 to 1.8 for LAI scaling and 0 to 2 for beta.

Reply to Q4-1: We thank the reviewer for summary of the main modelling findings and the relationships between LAI,  $\beta$ , and water partitioning. And yes, we agree that increasing LAI and decreasing  $\beta$  enhance evapotranspiration relative to recharge, and that the magnitudes of these effects depend on the chosen parameter ranges.

As already detailed in our reply to Q2 of Reviewer #1, the LAI scaling factors (0.2–1.8) were selected to represent canopy density variations from strongly thinned to dense stands within realistic limits of observed European forests. A similar scaling approach has been applied in previous tracer-aided modelling (e.g., (Neill et al., 2021). Reported maximum LAI values of up to 9.5 m² m² for mature beech forests in Central Germany (Leuschner et al., 2006) support that our selected range captures realistic canopy densities for managed Central European forests. For the root distribution parameter ( $\beta$ ), the range of 0–2 was derived from site-specific calibrations across the five vegetation types (broadleaf, conifer, agroforestry, grassland, and cropland). The posterior  $\beta$  distributions converge toward smaller values (median < 1) for forest sites, indicating deeper rooting compared with shallower-rooted agricultural systems. This confirms that a  $\beta$  range of 0–2 is physically realistic and suitable for representing root distribution across management scenarios.

We will expand Section 3.3 in the revised manuscript to clarify the derivation and justification of both the LAI and  $\beta$  parameter ranges.

Please also note that the variations in vegetation parameters—including forest-type-specific LAI, LAI scaling factors, and the root distribution parameter ( $\beta$ )—were selected in combination to represent the full spectrum of realistic land-use changes and forest management scenarios (e.g., differences in forest type, forest density, and rooting depth) for this geographical region, rather than purely theoretical model sensitivity tests of LAI or  $\beta$ . In particular,  $\beta$  captures variations in rooting depth from young to mature stands, reflecting forest age effects that are central to forest management. This aligns with the study's main objective of developing a parsimonious and generic forest management framework to evaluate how vegetation structure and rooting characteristics influence long-term water partitioning and ecohydrological resilience under comparable environmental conditions. While the qualitative effects of LAI and  $\beta$  can be analytically inferred from the model formulation, our scenario-based framework quantifies their magnitudes under physically constrained parameter ranges to assess vegetation structural effects on long-term water partitioning and ecohydrological resilience. However, in retrospect, we can see that we need to stress more that the modelling approach is more specific for regions similar soil/climatic conditions.

Q4-2. Alternatively, I suggest a stronger focus on dynamics introduced by wet/dry years, or when analysing monthly fluxes instead of longterm yearly averages would better justify the sensitivity analysis through carefully chosen synthetic applications of the dynamic model. Reply to Q4-2: We appreciate the reviewer's valuable suggestion to strengthen the investigation of temporal dynamics. While the current results primarily focus on long-term mean annual partitioning to isolate structural vegetation effects, the model is fully dynamic and resolves processes at a daily time step. In the revised manuscript, we will add a new and concise analysis in the Results section illustrating how evapotranspiration and recharge under different forest management scenarios respond to interannual (wet vs. dry years) and seasonal variability. Correspondingly, the Discussion (Section 5.2) will be expanded to interpret these dynamic responses and highlight how vegetation structure modulates hydroclimatic sensitivity across contrasting years.

**Minor suggestions:**

Q5. The structure of the manuscript should be thoroughly revised and streamlined to help the reader understand the study approach. It introduces concepts that are unnecessary to understand the results and discussion (e.g. mulching) or that are disregarded by the chosen methodology (e.g. effective calibration and equifinality, or dynamic, species-specific root distributions). Moreover, model calibrations to grassland, cropland (and agroforestry?), are not used except for Table 3 (and Table S2).

Reply to Q5: We thank the reviewer for this constructive comment. In the revised manuscript, we will streamline the Study Area and Methods sections to enhance clarity and focus on the elements directly relevant to the modeling framework and scenario analysis. Specifically, we will remove or condense non-essential information or concepts—for example, the brief mention of mulching in Section 2.1 (Study Area)—as this process is not relevant to our modeling framework and will be deleted.

We will also clarify the calibration procedure (see also response to Q9) by more clearly describing the two-step calibration process and explaining how the retained parameter sets were used for final simulations. To address the reviewer's concern about equifinality, we will include parameter probability density plots (PDFs) for all calibrated parameters at each real site to visualize the range and convergence.

Regarding the model calibrations for grassland, cropland, and agroforestry, this point is also addressed in response to Q6. As explained, the main aim of this study was not to build independently calibrated models for each site, but to develop a generic forest-management scenario framework. Site-specific calibrations (Table 3 and Table S2) were used to test model transferability and robustness, while the scenario experiments focused on varying vegetation-related parameters—mainly Leaf Area Index (LAI) and the root-distribution factor ( $\beta$ )—under consistent soil and climatic conditions. Importantly, as also noted in our reply to Q1, the revised version will explicitly describe how vegetation parameters (such as rE,  $\alpha$ , and INTp) were transferred from the calibrated site models (broadleaf, conifer, and agroforestry) into the generic framework to ensure physical consistency as well as transparency and reproducibility.

Q6. It is unclear whether the model fitted to agroforestry has been used anywhere else than in Table 3. Please clarify. Also note that Table S2 indicates the BF model to have a calibrated Lmax parameter that differs from the AF model. This finding additionally corroborates the invalidity of the extrapolation approach mentioned earlier in this review.

Reply to Q6: We thank the reviewer for this comment. In the previous version, the model fitted to the agroforestry (AF) site was indeed only shown in Table 3. As mentioned above, in the

revised manuscript, we will use the vegetation parameters (rE,  $\alpha$ , INTp) derived from all calibrated forest sites (broadleaf, conifer, and agroforestry) in the generic framework.

As mentioned in reply to Q2, we acknowledge that Table S2 in the previous submission mistakenly presented the calibrated ranges as initial parameter ranges. We provide the corrected version of Table S2 in this reply report for clarity and reference. The slight differences in the calibrated Lmax values between the broadleaf forest (BF) and agroforestry (AF) sites are realistic given site-specific soil conditions and do not indicate an error. Instead, they reinforce the rationale for keeping soil parameters unchanged and focusing on vegetation changes within this generic scenario framework.

- Q7. I suggest to improve the focus on the minimum of results needed to support the findings, instead of representing the model output in various forms. Some Figures and Tables are unclear (e.g Table 1) or redundant (e.g. Fig 7/8 or Fig. 4d = sum of 4b/c) and should be reconsidered. Also consistent color schemes (e.g. throughout Figures 3,5,6,11) would help the reader. Reply to Q7: We thank the reviewer for this valuable comment and fully agree that the presentation of results can be streamlined much more to improve focus and readability. To address these points, we will revise and simplify the figures and tables accordingly, as detailed below.
- (a) Table 1 summarizes soil properties and soil moisture statistics at the broadleaf forest site. To reduce redundancy, it will be moved to the Supplementary Material and extended to include data for all monitored forest sites (e.g., broadleaf, conifer forest, and agroforestry).
- (b) Figures 7 and 8 describe ecohydrological responses across forest types and management scenarios in annual mean form. Figure 7 presents the full-matrix (heatmap) visualization, whereas Figure 8 shows the same relationships as sensitivity curves. To simplify the main text, we will retain Figure 8 in the main manuscript and move the Figure 7 to the Supplementary Material.
- (c) Since panel Figure 4d represents the sum of 4b and 4c, we will delete panels 4b and 4c and keep 4d as the main summary figure.
- (d) We will unify the color palette across all figures (e.g., Figs. 3, 5, 6, 11) to ensure consistent representation of forest types and water flux components. In line with the editor's guidance (Mario Ebel), all revised figures will also be checked for accessibility to readers with color-vision deficiencies using the Coblis Color Blindness Simulator.

Revised Table 1. Summary of observed soil types and soil moisture data at the three forest sites.

| Site                | C - :1 T                                | Texture            | Layer        | Soil Moisture (mm) |       |        |       | Period             |
|---------------------|-----------------------------------------|--------------------|--------------|--------------------|-------|--------|-------|--------------------|
| Site                | Soil Type                               |                    |              | Max                | Min   | Mean   | SD    |                    |
| Broadleaf
forest | Brown Earth                             | Loamy
sand/sand | 0 to 10 cm   | 26.28              | 3.50  | 13.67  | 6.30  | 2018.6-
2024.12 |
|                     |                                         |                    | 10 to 30 cm  | 56.19              | 6.86  | 24.68  | 11.70 |                    |
|                     |                                         |                    | 30 to 100 cm | 147.51             | 25.83 | 71.71  | 33.50 |                    |
|                     | Gley (Sand)                             | Sand,
compacted | 0 to 10 cm   | 28.65              | 8.62  | 17.32  | 7.07  |                    |
| Conifer forest      |                                         |                    | 10 to 30 cm  | 53.75              | 2.59  | 21.78  | 12.29 |                    |
|                     |                                         |                    | 30 to 100 cm | 34.70              | 2.68  | 15.85  | 7.96  | 2019.3-            |
| Agroforestry        | Podsolic Loamy
Brown Earth sand/sand |                    | 0 to 10 cm   | 32.06              | 10.40 | 21.25  | 7.77  | 2024.12            |
|                     |                                         | Loamy
sand/sand | 10 to 30 cm  | 53.35              | 7.15  | 29.75  | 13.49 |                    |
|                     |                                         |                    | 30 to 100 cm | 223.62             | 86.83 | 163.41 | 41.98 |                    |

Q8. If available, the use of d18O in combination with d2H might help to distinguish evaporation from mixing effects (e.g. Penna 2018) and thus improve model calibration. Reply to Q8: We thank the reviewer for this valuable suggestion. We agree that combining  $\delta^{18}$ O and  $\delta^{2}$ H can better distinguish evaporation from mixing effects (Penna et al., 2018) and thus has the potential to improve model calibration. In this study, however, we used  $\delta^{2}$ H only, following recent isotope-aided ecohydrological modelling applications in this region (e.g. Landgraf et al., 2023), where  $\delta^{2}$ H provided sufficient sensitivity to evaporative fractionation and avoided potential carbonate-related biases that can affect  $\delta^{18}$ O in soil waters (Meißner et al., 2014). Nonetheless, we acknowledge the value of dual-isotope calibration and plan to explore this approach in future EcoPlot-iso developments. We will also clarify this rationale and limitation in the Discussion section of the revised manuscript.

Q9. The provided description of methodology is not sufficient for reproduction, e.g. how are rL1, rL2, rl3 linked to Eq.4, how was the model re-run with the "retained parameter space", is recharge defined at the lower boundary of the simulation domain (how was the size of the domain defined and does it affect timing of the fluxes e.g. in Figure 4)?

Reply to Q9: We thank the reviewer for this valuable comment and apologise for not being clearer. Below, we provide point-by-point clarifications how we will address all mentioned issues and how we will revise the Methods section accordingly to improve reproducibility.

**(1) Definition of rL1–rL3 and linkage to Eq. (4):**

As described in Section 3.2, r(z) represents the depth-dependent root water withdrawal efficiency at depth z (Eq. 4). The model domain is divided into three soil layers (0–10 cm, 10–30 cm, and 30–100 cm; Fig. 2a). For each layer, we use the midpoint depth (5, 20, and 65 cm) to calculate the corresponding root water uptake efficiencies as rL1 = r(5 cm), rL2 = r(20 cm), and rL3 = r(65 cm). These layer-specific efficiency factors are then used as coefficients in Eqs. (1)–(3) to represent the vertical distribution of root water uptake capacity across the three layers. We will explicitly state this (mid-point-depth) linkage in Section 3.2 of the revised manuscript.

(2) Re-running the model with the retained parameter space:

As described in Section 3.4, 100,000 parameter sets were initially generated using Latin Hypercube Sampling within a Monte Carlo framework to explore the full parameter space. Each simulation covered 25 years and produced 27 output variables, so only the modified Kling–Gupta Efficiency (mKGE) values and the associated parameter sets were stored to reduce data volume. Based on these results, we retained parameter sets that fell within the 60th-percentile intersection of the multi-criteria mKGE (averaged across soil moisture and soil-water isotope metrics at all three depths).

The model was then re-run using these retained parameter sets to generate complete simulations and refine parameter estimates. From this refined ensemble, the 100 best-performing runs (with the highest averaged mKGE values across the three soil layers) were selected for final analysis. In the revised manuscript, we will clarify this two-step calibration procedure and explicitly describe how the 60th-percentile intersection was used to select and re-run parameter sets in Section 3.4 of the revised manuscript.

**(3) Definition of groundwater recharge and model domain:**

As illustrated in Figure 2, groundwater recharge in EcoPlot-iso is defined at the lower boundary of the 1 m soil domain as the downward percolation flux from the deepest soil layer (30–100 cm) to the groundwater. (a) This 1 m depth is consistent with previous EcoPlot-iso applications (Birkel et al., 2024; Landgraf et al., 2023; Stevenson et al., 2023), where most soil–plant–atmosphere interactions occur within the upper meter of soil. (b) It also corresponds to the range of in situ soil moisture and isotope sensors at our study sites. (c) In addition, field observations show that groundwater tables are relatively shallow (0–4 m) in this lowland catchment.

Although testing deeper domains was beyond the scope of this study, we acknowledge that extending the lower boundary beyond 1 m would increase soil water storage and delay drainage, potentially affecting recharge timing at sub-daily or daily timescales. However, when fluxes are aggregated at the monthly scale (as in Figure 4), these timing differences become negligible and the overall water balance remains largely unaffected. Therefore, the 1 m domain provides a physically justified and widely used approximation for representing recharge and evapotranspiration processes in lowland catchments.

We will make this rationale explicit in Section 3.1 of the revised manuscript, clarifying the choice of the 1 m domain and its implications for recharge timing.

**References:**

- Birkel, C., Arciniega-Esparza, S., Maneta, M. P., Boll, J., Stevenson, J. L., Benegas-Negri, L., Tetzlaff, D., & Soulsby, C. (2024). Importance of measured transpiration fluxes for modelled ecohydrological partitioning in a tropical agroforestry system. *Agricultural and Forest Meteorology*, 346. https://doi.org/10.1016/j.agrformet.2023.109870
- Landgraf, J., Tetzlaff, D., Birkel, C., Stevenson, J. L., & Soulsby, C. (2023). Assessing land use effects on ecohydrological partitioning in the critical zone through isotope-aided modelling. *Earth Surface Processes and Landforms*, 48(15), 3199–3219. https://doi.org/10.1002/esp.5691
- Leuschner, C., Voß, S., Foetzki, A., & Clases, Y. (2006). Variation in leaf area index and stand leaf mass of European beech across gradients of soil acidity and precipitation. *Plant Ecology*, *186*(2). https://doi.org/10.1007/s11258-006-9127-2
- Meißner, M., Köhler, M., Schwendenmann, L., Hölscher, D., & Dyckmans, J. (2014). Soil water uptake by trees using water stable isotopes ( $\delta$ 2H and  $\delta$ 18O)—a method test regarding soil moisture, texture and carbonate. *Plant and Soil*, 376(1-2). https://doi.org/10.1007/s11104-013-1970-z
- Neill, A. J., Birkel, C., Maneta, M. P., Tetzlaff, D., & Soulsby, C. (2021). Structural changes to forests during regeneration affect water flux partitioning, water ages and hydrological connectivity:

- Insights from tracer-aided ecohydrological modelling. *Hydrology and Earth System Sciences*, 25(9), 4861–4886. https://doi.org/10.5194/hess-25-4861-2021
- Penna, D., Hopp, L., Scandellari, F., Allen, S. T., Benettin, P., Beyer, M., Geris, J., Klaus, J., Marshall, J. D., Schwendenmann, L., Volkmann, T. H. M., Von Freyberg, J., Amin, A., Ceperley, N., Engel, M., Frentress, J., Giambastiani, Y., McDonnell, J. J., Zuecco, G., ... Kirchner, J. W. (2018). Ideas and perspectives: Tracing terrestrial ecosystem water fluxes using hydrogen and oxygen stable isotopes Challenges and opportunities from an interdisciplinary perspective. *Biogeosciences*, 15(21). https://doi.org/10.5194/bg-15-6399-2018
- Stevenson, J. L., Birkel, C., Comte, J. C., Tetzlaff, D., Marx, C., Neill, A., Maneta, M., Boll, J., & Soulsby, C. (2023). Quantifying heterogeneity in ecohydrological partitioning in urban green spaces through the integration of empirical and modelling approaches. *Environmental Monitoring and Assessment*, 195(4). https://doi.org/10.1007/s10661-023-11055-6